# *Helicobacter pylori* induces the expression of Lgr5 and stem cell properties in gastric target cells

Zuzana Nascakova[1,*], Jiazhuo He[1,*], Giovanni Papa[1,*], Biel Francas[1], Flora Azizi[1], Anne Müller[1,2]

*Helicobacter pylori* infection predisposes carriers to a high risk of developing gastric cancer. The cell-of-origin of antral gastric cancer is the Lgr5[+] stem cell. Here, we show that infection of antrum-derived gastric organoid cells with *H. pylori* increases the expression of the stem cell marker Lgr5 as determined by immunofluorescence microscopy, qRT-PCR, and Western blotting, both when cells are grown and infected as monolayers and when cells are exposed to *H. pylori* in 3D structures. *H. pylori* exposure increases stemness properties as determined by spheroid formation assay. Lgr5 expression and the acquisition of stemness depend on a functional type IV secretion system (T4SS) and at least partly on the T4SS effector CagA. The pharmacological inhibition or genetic ablation of NF-κB reverses the increase in Lgr5 and spheroid formation. Constitutively active Wnt/β-catenin signaling because of *Apc* inactivation exacerbates *H. pylori*–induced Lgr5 expression and stemness, both of which persist even after eradication of the infection. The combined data indicate that *H. pylori* has stemness-inducing properties that depend on its ability to activate NF-κB signaling.

## Introduction

Colonization of the human gastrointestinal tract with *Helicobacter pylori* (*H. pylori*) or with genotoxic strains of *E. coli* or *Salmonella enterica* is directly linked to cancers of the stomach, the colon, and the gallbladder (Nagaraja & Eslick, 2014; Amieva & Peek, 2016; Mughini-Gras et al, 2018; Dziubanska-Kusibab et al, 2020; Pleguezuelos-Manzano et al, 2020). The causal link to cancer is especially well documented for *H. pylori* and gastric cancer (Plummer et al, 2015), the fifth most common cause of cancer-related deaths (Bray et al, 2024). *H. pylori* infection affects roughly half of the world's population but is asymptomatic in most individuals; ~1% of the *H. pylori*–infected population will develop gastric cancer in the course of their lifetimes. *H. pylori* is particularly detrimental in carriers of inherited germline mutations affecting genes whose protein products are involved in the repair of DNA double-strand breaks (DSBs) by homologous recombination (HR) repair (Usui et al, 2023). Carriers of pathogenic variants in the HR genes *BRCA1*, *BRCA2*, *ATM*, and *PALB2* exhibit a strongly elevated risk of developing gastric cancer but only if they are infected with *H. pylori* (Usui et al, 2023). The observed synergy between *H. pylori* infection and HR deficiency in gastric carcinogenesis appears to be attributable to the ability of *H. pylori* to induce DNA DSBs in its target cells (Toller et al, 2011; Hanada et al, 2014; Hartung et al, 2015; Bauer et al, 2020; Imai et al, 2021; He et al, 2023), as these DNA lesions can only be repaired in an error-free manner if HR repair is functional. In settings of HR deficiency, cells are forced to resort to error-prone repair pathways such as non-homologous end-joining and microhomology-directed repair, which result in the accumulation of mutations and, ultimately, malignant transformation (Muller & He, 2023; Usui et al, 2023). *H. pylori*–induced DNA damage occurs in proliferating cells in S-phase and depends on a functional type IV secretion system (T4SS) and the cytoplasmic delivery of the T4SS effector β-ADP heptose to the target cell cytosol (Bauer et al, 2020; He et al, 2023). Cytoplasmic β-ADP heptose binds to and activates the innate immune sensor alpha-kinase 1 (ALPK1), which in turn activates NF-κB and the subsequent expression of pro-inflammatory cytokines and other NF-κB target gene products (Gall et al, 2017; Stein et al, 2017; Zimmermann et al, 2017). Experimental infection of mice with T4SS-proficient *H. pylori* has shown that, at least in the antral (distal) part of the stomach, gland base stem cells are the predominant cell lineage responding to *H. pylori* and its β-ADP heptose with the production of pro-inflammatory mediators (Wizenty et al, 2022). The expression and secretion of antimicrobial factors by gland-base cells are NF-κB dependent (Wizenty et al, 2022). Interestingly, the antimicrobial response of Lgr5[+] antral gland base cells results in the efficient control of the bacteria and presumably serves to protect the gland base and its stem cell pool (Sigal et al, 2019). Complementing these various findings, we have identified Lgr5[+] antral stem and progenitor cells, and their Troy[+] gastric corpus counterparts, as the preferred target of *H. pylori*–induced DNA damage in vitro using organoid cells exposed to T4SS-proficient *H. pylori* (He et al, 2023).

[1]Institute of Molecular Cancer Research, University of Zürich, Zürich, Switzerland [2]Comprehensive Cancer Center Zürich, Zürich, Switzerland

Correspondence: mueller@imcr.uzh.ch
*Zuzana Nascakova, Jiazhuo He, and Giovanni Papa contributed equally to this work

Antral gastric stem cells are marked by Lgr5 expression, a trait that is shared with small intestinal, colonic, and hair follicular stem cells (Barker et al, 2007, 2008, 2010), and stem cell populations of other highly regenerative organs such as the liver (Huch et al, 2013). Lgr5⁺ stem cells drive cell renewal and gland regeneration in the antral stomach at steady state (Barker et al, 2010). Lgr5⁺ stem cells have more recently also been identified in the gastric corpus, where they reside at the gland base and appear to arise from differentiated chief cells in settings of tissue damage and the ensuing repair (Leushacke et al, 2017). Corpus gland base cells thus serve as reserve stem cells; they are marked by Lgr5 and, in addition, by Runx1, Mist1, and Troy (Leushacke et al, 2017). Gastric Lgr5⁺ stem cells not only regenerate gastric glands at steady state and after tissue damage, respectively, but also, upon oncogenic insult, may serve as cells of origin of advanced gastric cancer in mice; such experimentally induced gastric cancers, or their precursor lesion "spasmolytic polypeptide-expressing metaplasia" (SPEM), can be driven by $Kras^{G12D}$ expression alone (Leushacke et al, 2017) or in conjunction with Apc and Trp53 ablation (Fatehullah et al, 2021). In vivo ablation of the tumor-resident Lgr5⁺ cancer stem cell pool has revealed a role for these cells in cancer initiation and maintenance, as well as metastasis (Fatehullah et al, 2021). Pten and Smad4 ablation in Lgr5⁺ murine gastric stem cells was also independently shown to give rise to gastric cancer of the intestinal type (Li et al, 2016). LGR5 expression is a hallmark not only of murine but also of human gastric cancers (Leushacke et al, 2017). Lgr5 marks cancer stem cells that give rise to both corpus and antrum/pyloric gastric cancer in mice (Leushacke et al, 2017; Tan et al, 2020; Fatehullah et al, 2021), recapitulating the dominant gastric sites at which human non-cardia gastric cancer arises (Kim & Choi, 2019).

Given that Lgr5⁺ stem cells give rise to experimentally induced gastric cancer in vivo and are targeted by H. pylori both in vivo (Wizenty et al, 2022) and in vitro (He et al, 2023), we asked how H. pylori would affect the biology of this lineage. We set out to investigate, using organoid cells, whether and how H. pylori would increase the stem cell pool and induce stem cell properties in its target cells. Infection of mice and the immunohistochemical evaluation of human biopsies had previously suggested that the stem and progenitor compartment expands during H. pylori infection (Uehara et al, 2013; Sigal et al, 2015). We show here, using organoid cells infected with H. pylori in 2D or 3D, that H. pylori induces the stem cell marker Lgr5 in its target cells and that exposure to T4SS-proficient, but non-deficient H. pylori results in enhanced stemness properties. Hyperactivation of the Wnt signaling pathway by inactivation of the tumor suppressor Apc exacerbated the H. pylori–induced stemness of target cells. Lgr5 induction and stemness could further be prevented completely by the pharmacological inhibition or genetic inactivation of NF-κB and was stable even after antibiotic eradication of the bacteria.

## Results

### H. pylori induces Lgr5 expression in antral murine organoid cells upon 2D or 3D infection

To assess whether H. pylori infection affects Lgr5 expression, we cultured organoids from antrum or corpus tissue harvested from C57BL/6 mice, seeded the organoid cells in 2D and exposed them to the H. pylori strains PMSS1 or G27 for 6 h; both strains harbor a functional T4SS and adhere to murine cells in culture (He et al, 2023). Exposure to G27, and to a lesser extent PMSS1, resulted in enhanced Lgr5 expression as determined by immunofluorescence microscopy using an antibody clone that had been validated for this application using organoids from an Lgr5-mOrange reporter mouse (He et al, 2023), followed by automated signal quantification (Fig 1A–D); both the mean intensity of the Lgr5 signal per cell and the frequency of Lgr5⁺ cells increased because of infection (Fig 1A–D). Lgr5 expression increased in a multiplicity of infection (MOI)-dependent manner; as an MOI of 50 showed strong effects on Lgr5 expression without completely overgrowing the cells or causing cytotoxicity (Fig S1A, and data not shown), this MOI was used throughout the study unless otherwise indicated. H. pylori was found to adhere to both Lgr5⁺ and Lgr5⁻ cells in the culture (Fig S1A). An increase in the expression of Lgr5 could also be detected at the transcript level by qRT-PCR (Fig 1E). Western blotting further also confirmed the up-regulation of Lgr5, not only in antrum-derived cultures (Fig 1F and G) but also in corpus-derived cultures in which the Lgr5 signal was generally weaker (Fig S1B). The induction of Lgr5 was fairly specific, as other gastric stem cell markers, such as Sox2, Troy, Runx1, Lrig1, and others (Liabeuf et al, 2022) were not induced as a consequence of infection (Fig S1C); an exception was Ascl2, the expression of which mirrored that of Lgr5 (Fig S1C). Infection of organoid cells generated from a transgenic Lgr5-mOrange reporter mouse line (Fazilaty et al, 2021; Reichmuth et al, 2021 Preprint) confirmed the up-regulation of Lgr5-driven mOrange expression (Fig 1H–J) and confirmed that Lgr5 expression is induced by H. pylori at the transcriptional level. To address whether H. pylori also induces Lgr5 expression in organoids growing in Matrigel as 3D structures, we microinjected the bacteria into organoids and maintained injected and non-injected organoids for another 6 h before performing immunofluorescence microscopy; this approach confirmed that bacteria are indeed detectable in the lumen and that Lgr5 expression is higher in the H. pylori–exposed organoids than in controls (Fig 1K and L; high-magnification images in Fig S1D). We next developed a less time-consuming and more scalable approach to encapsulating the bacteria inside organoids (Fig S1E); immunofluorescence microscopy and Lgr5-specific Western blotting of such H. pylori–encapsulated organoids confirmed that Lgr5 expression was higher in organoids co-cultured with H. pylori than in controls (Figs 1M and N and S1F). The combined results indicate that H. pylori induces the stem cell marker Lgr5 in its target cells under both 2D and 3D co-culture conditions.

### H. pylori–induced Lgr5 expression depends on a functional T4SS and NF-κB activation

We next asked whether the H. pylori T4SS and the T4SS-secreted effectors CagA and/or β-ADP heptose were required for Lgr5 induction. In the G27 strain background, a mutant lacking the entire Cag pathogenicity island encoding the T4SS was completely defective for Lgr5 induction as determined by microscopy of cells infected in 2D, both with respect to the mean Lgr5 intensity per cell and the frequency of Lgr5⁺ cells (Fig 2A–D). A mutant lacking only CagA showed an intermediate phenotype and had some residual

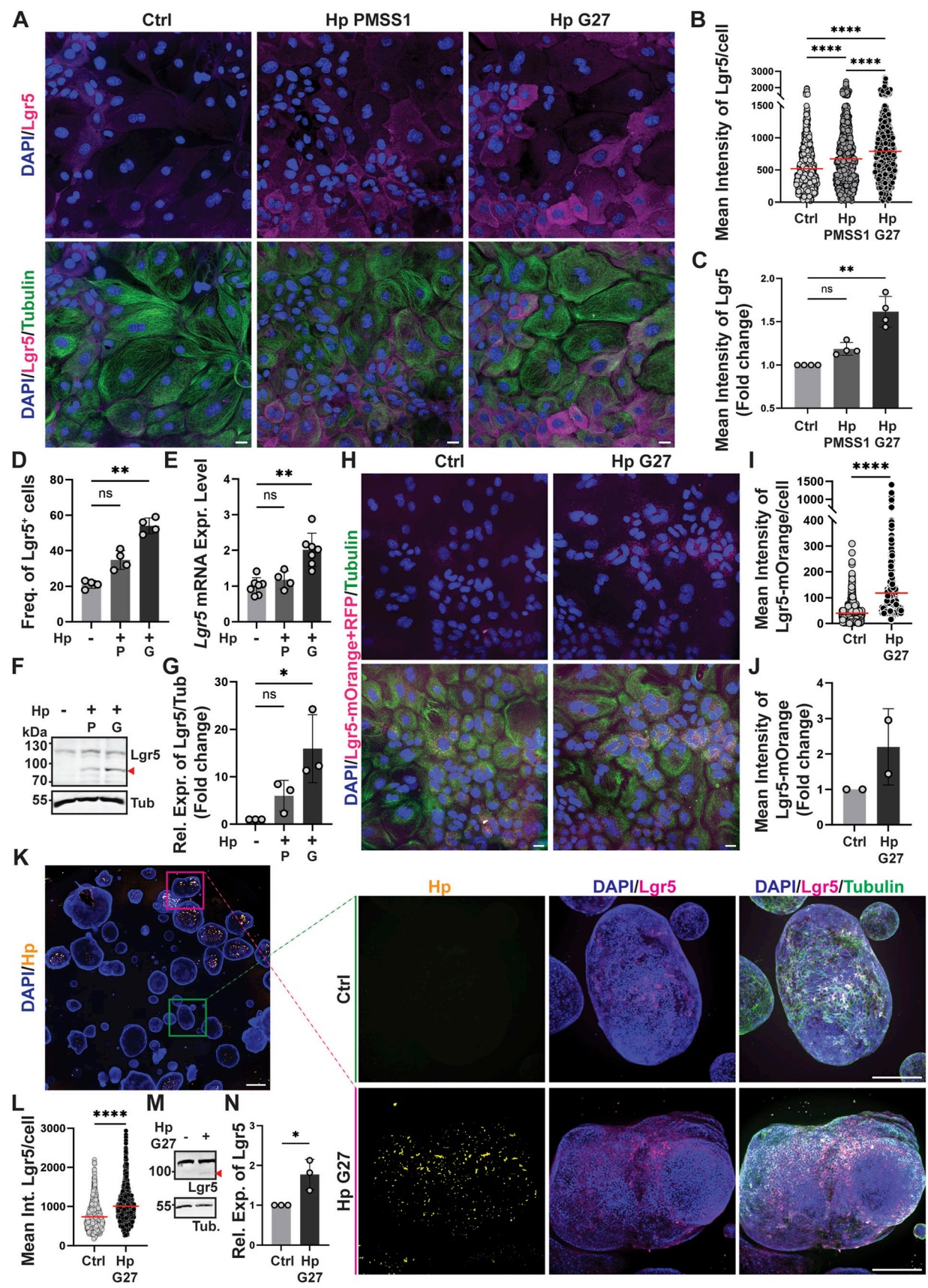

Lgr5-inducing activity (Fig 2A–D). The same was true for the PMSS1 strain background, in which T4SS deficiency (in the ΔCagE mutant) completely abolished the strain's Lgr5-inducing activity, whereas lack of CagA again showed an intermediate phenotype (Fig 2E–H). This could be confirmed for G27 also at the transcript level (Fig 2I) and by Western blotting (Fig 2J and K). A PMSS1 mutant lacking the ability to produce and deliver the LPS biosynthetic intermediate β-ADP heptose (ΔRfaE) was completely deficient for Lgr5 induction (Fig S2A–C). Addition of β-ADP heptose alone was not sufficient to induce Lgr5 (Fig S2A–C), also not when combined with a ΔPAI mutant infection (Fig S2D–F). As *H. pylori* T4SS activity results in NF-κB activation (Gall et al, 2017; Stein et al, 2017; Zimmermann et al, 2017), we asked whether NF-κB inhibition would abrogate Lgr5 induction. This was indeed the case for both strains using an inhibitor of the IκBα kinase IKK, which prevents IκBα phosphorylation and the nuclear translocation of NF-κB; exposure to this compound reduced both the mean Lgr5 intensity per cell, the frequency of Lgr5$^+$ cells as determined by microscopy (Fig S2G–J), and Lgr5 expression as determined by Western blotting (Fig S2K and L). The combined results support the idea that NF-κB activation downstream of the *H. pylori* T4SS/target cell interaction drives Lgr5 expression.

### *H. pylori* synergizes with constitutively active Wnt signaling to induce Lgr5 expression in 2D and 3D-cultured organoid cells

*Lgr5* is a known target gene of the Wnt signaling pathway (Barker et al, 2007). The dysregulation of Wnt signaling resulting from the mutational inactivation of Wnt pathway components is a well-known feature of human gastric cancer (Koushyar et al, 2020). Dysregulated Wnt signaling affects almost 50% of gastric cancers (Ooi et al, 2009), with the most commonly mutated genes being *RNF43*, *AXIN1/2*, *CTNNB1*, and *APC*. We asked whether *H. pylori* infection would synergize with *Apc* inactivation to induce Lgr5 expression in organoid cells. Organoid cells were cultured from WT and *Apc*$^{min/+}$ mice, in which one allele encodes a stop codon at position 850 leading to premature truncation of the protein (Moser et al, 1990), and were infected with *H. pylori* strain G27 before staining for Lgr5. *Apc*$^{min/+}$ organoid cells expressed more Lgr5 at a

steady state than their WT counterparts, and this could be further increased by infection (Fig 3A–C). Western blotting confirmed the additive effects of *Apc* truncation and infection in organoid cells grown in 2D (Fig 3D and E). Encapsulation of bacteria into 3D organoids, followed by immunofluorescence microscopy or Western blotting, further confirmed the additive effects of the *Apc* truncation and infection also for 3D organoid cultures (Fig 3F–H). The combined results suggest that gastric cells with a pre-existing hyperactivation of the Wnt signaling pathway respond more strongly to *H. pylori* infection, which manifests in higher expression of Lgr5.

### *H. pylori* and constitutively active Wnt signaling induce stem cell activity as determined by spheroid formation assay

To examine whether the *H. pylori*–induced increase in Lgr5 expression is accompanied by elevated stem cell activity, we subjected organoid cells to a spheroid formation assay (SFA), for which organoid cells are grown on cell-repelling surfaces that favor 3D growth in the form of small spheres or spheroids. This assay was first developed for prostate and breast stem cells (Dontu et al, 2003; Lawson et al, 2007; Xin et al, 2007; Garraway et al, 2010). Only undifferentiated stem cells with self-renewing capacity are capable of forming spheroids/spheres when grown on special cell-repelling surfaces; therefore, the SFA is an accepted method to quantitatively assess stem cell properties (Pastrana et al, 2011). Organoid cells grown as 2D monolayers and infected with *H. pylori* failed to form spheroids (data not shown); therefore, we resorted to 3D growth and *H. pylori* encapsulation for the SFA. In brief, organoids were broken up into small pieces, which were incubated for 6 h in the presence of *H. pylori* at a relatively high calculated MOI of 250. After six hours of encapsulation, single-cell suspensions were generated and seeded at various starting densities into cell-repelling Biofloat plates and allowed to form spheres for up to 7 d; a seeding density of 8,000 cells per well, and an optimal time point of 5 d were chosen based on pilot experiments (Fig S3A–C) for further experimentation. Exposure of 3D-grown organoid cells to encapsulated *H. pylori* G27 or PMSS1

---

**Figure 1. H. pylori infection increases the expression of the stem cell marker Lgr5 in murine gastric antrum-derived organoids grown as monolayers or in 3D.**
**(A, B, C, D, E, F, G)** Antrum-derived WT organoid cells seeded as monolayers in 2D culture medium were infected with *H. pylori* strains PMSS1 or G27 (MOI of 50) for 6 h. Lgr5 expression was assessed by quantitative immunofluorescence (IF) microscopy, Western blotting and qRT-PCR. **(A, B)** Representative IF images are shown in (A) (Lgr5 in magenta, β-tubulin in green, scale bar, 20 μm); the quantification of Lgr5 expression of a representative experiment is shown in (B) (each symbol represents one cell, i.e., the mean intensity of all the pixels of that cell; red lines indicate means). **(C)** The mean Lgr5 intensities of four independently conducted experiments are plotted as mean ± SD of the fold change over (uninfected) control in (C). **(C, D)** The frequencies of Lgr5$^+$ cells (i.e., with a mean intensity > the mean of the control condition), among all cells, of the four experiments in (C) are shown in (D); symbols in (C, D) correspond to independent experiments/biological replicates. **(E)** Lgr5 transcripts, normalized to Gapdh, are quantified for four to seven experiments in (E) and plotted as mean ± SD. **(F, G)** A representative Western blot is presented in (F) (Lgr5, ~100 kD, indicated by red arrows, and α-tubulin, ~50 kD) alongside the quantification of three independent experiments (plotted as mean ± SD, normalized to tubulin and the control condition) in (G). P indicates *H. pylori* strain PMSS1; G indicates G27. **(H, I, J)** Lgr5-2A-CreERT2-2A-mOrange2 mice were used to generate gastric antrum-derived organoids. Organoid cells were seeded as monolayers and infected with *H. pylori* G27 (MOI of 50) for 6 h and stained for Lgr5-mOrange (anti-RFP, in magenta) and β-tubulin (in green). **(H, I, J)** Representative images are shown in (H) (scale bar, 20 μm) alongside the Lgr5 quantification of a representative experiment (in I) and of two independent experiments (in J) as described above. **(K, L)** Antrum-derived gastric organoids growing in Matrigel were infected with *H. pylori* strain G27 by microinjection for 6 h. 3D organoids were stained for Lgr5 (in magenta), *H. pylori* (in yellow), and β-tubulin (in green), counterstained with DAPI, and imaged by Confocal spinning disk microscopy. A low-magnification image of a Matrigel drop is shown in the left panel (scale bar, 500 μm). High-magnification images of two selected organoids with or without *H. pylori* are shown in the right panel (scale bar, 200 μm). **(L)** The images of 3D organoids were rendered using Fiji software and Maximum Intensity Z-Projection and their Lgr5 signal quantified (in L) using CellProfiler. **(M, N)** Antrum-derived 3D organoids were infected with *H. pylori* strain G27 (MOI of 50) by encapsulation for 6 h and subjected to Western blotting of protein extracts. **(M, N)** A representative blot is shown in (M) alongside the quantification of three independent experiments in (N). **(B, C, D, I, J, L)** Data information: 600–1,000 cells per condition were analyzed in (B, C, D, I, J, L). **(B, C, D, E, G, I, J, L, N)** P-values were calculated using *t* test (two groups) or one-way ANOVA (more than two groups; non-parametric in (B, I, L); ordinary one-way ANOVA with Tukey's multiple comparisons test in (C, D, E, G, J, N)) and are indicated in ranges; *P < 0.05; **P < 0.01; ****P < 0.001; ns, not significant.

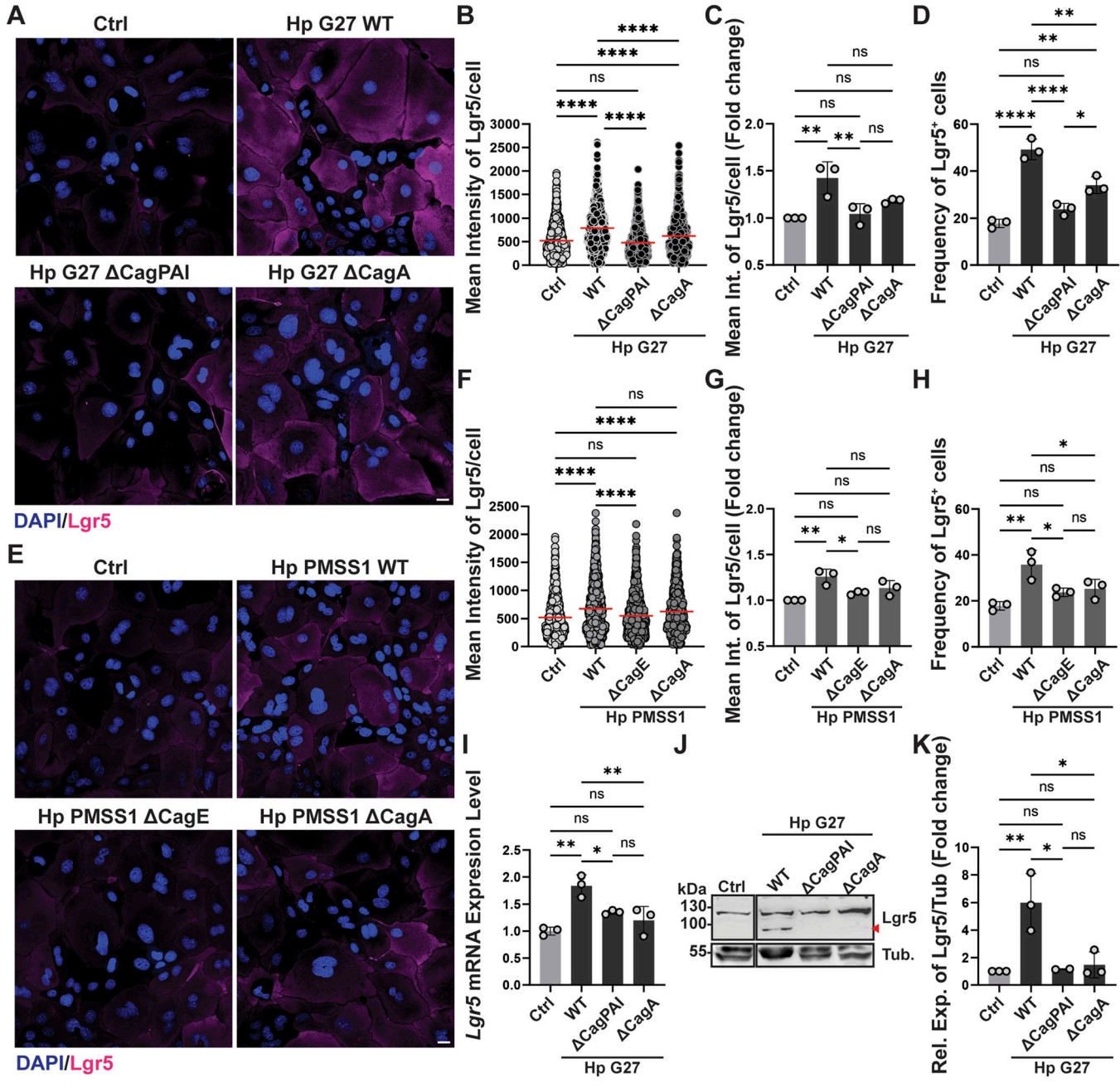

**Figure 2. *H. pylori*–induced Lgr5 expression depends on a functional T4SS and active NF-κB signaling.**
**(A, B, C, D, E, F, G, H, I, J, K)** Antrum-derived organoid cells were grown as monolayers and infected with WT *H. pylori* or the indicated isogenic mutants of strains G27 (A, B, C, D) or PMSS1 (E, F, G, H) at an MOI of 50 for 6 h. Lgr5 expression was assessed by quantitative IF microscopy (both strains) and by qRT-PCR and Western blotting (G27 only). **(A, E)** Representative IF images are shown in (A, E) (Lgr5 in magenta, DAPI in blue; scale bar, 20 μm). **(B, C, F, G)** The quantification of Lgr5 expression of a representative experiment is shown in (B, F); means ± SD of three independent experiments are shown in (C, G). **(D, H)** Frequencies of Lgr5⁺ cells among all cells are shown in (D, H). Symbols in (C, D, G, H) correspond to independent experiments. **(I)** Lgr5 transcripts, normalized to Gapdh, are quantified for three experiments in (I) and plotted as mean ± SD. **(J, K)** A representative Western blot is presented in (J) alongside the quantification of two (ΔPAI) or three (WT, ΔCagA) experiments in (K). **(B, C, D, F, G, H)** Data information: ~1,000 cells per condition were analyzed in (B, C, D, F, G, H). **(B, C, D, F, G, H, I, K)** P-values were calculated using one-way ANOVA (non-parametric in (B, F); ordinary one-way ANOVA with Tukey's multiple comparisons test in (C, D, G, H, I, K)); P-values are indicated in ranges; *P <0.05; **P < 0.01; ***P < 0.005; ****P < 0.001; ns, not significant.

strongly enhanced their ability to form spheroids (Fig 4A and B). The ability to induce spheroid formation was dependent on a functional T4SS and CagA in G27 (Fig 4C and D); similar trends were observed with PMSS1, although the number of spheroids forming was generally much lower (Fig S3D and E). NF-κB inhibition prevented the increase in spheroid formation induced by strain G27 (Fig S3F). *Apc* truncation exacerbated *H. pylori*–induced sphere formation (Fig 4E and F). The combined results suggest

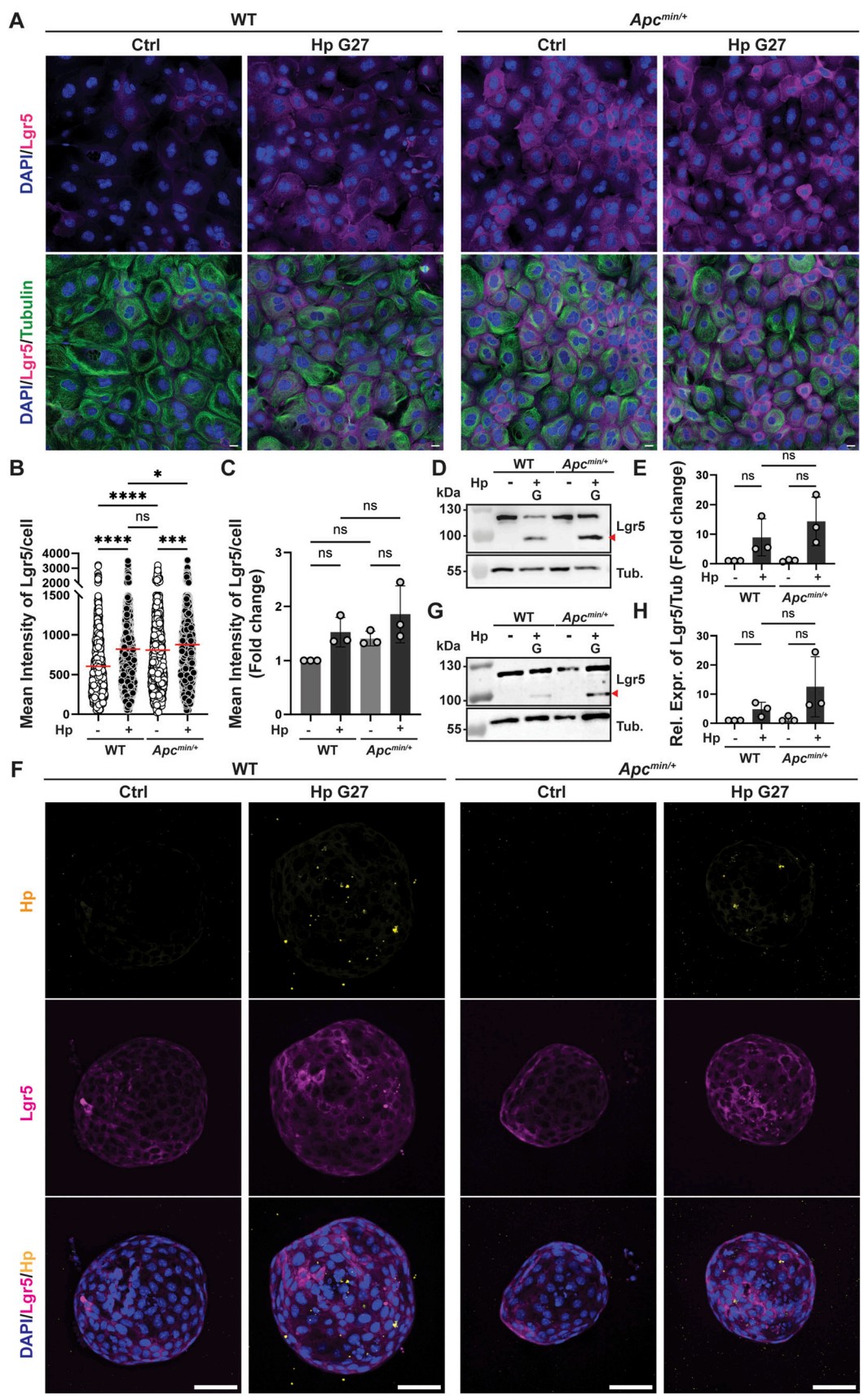

that *H. pylori* induces the stem cell marker Lgr5 and stem cell properties in its target cells in a T4SS- and NF-κB-dependent manner.

### RNAi-driven knockdown of the NF-κB subunit P65 reverses the effects of *H. pylori* infection on Lgr5 expression and stemness

To confirm the critical role of NF-κB in *H. pylori*–induced Lgr5 expression and stemness, we generated organoid cells with a constitutive overexpression of an shRNA targeting the NF-κB subunit p65. The knockdown of p65 was >75% as judged by qRT-PCR (Fig 5A) and *H. pylori*–induced NF-κB target gene expression (of *Cxcl1*, *Cxcl2*, and *Ccl2*) was strongly reduced (Fig 5B). Lgr5-specific immunofluorescence microscopy revealed that the p65-specific shRNA, but not the scrambled control shRNA, prevented Lgr5 induction by strain G27 (Fig 5C–E). This was confirmed by Lgr5-specific qRT-PCR (Fig 5F) and Western blotting (Fig 5G and H). Loss of p65 also compromised the ability of *H. pylori* G27 to induce spheroid formation; in fact, p65 knockdown cells had a much lower capacity to form spheroids already at baseline (Fig 5I and J), despite similar growth kinetics of the organoids. The combined results suggest that *H. pylori*–induced Lgr5 expression and enhanced stemness are dependent on NF-κB.

### The effects of *H. pylori* on Lgr5 expression and stem cell activity as determined by spheroid formation assay persist after antibiotic eradication of the bacteria

We next asked how dependent the enhanced Lgr5 expression and stem cell activity would be on continuous exposure to live bacteria. To this end, we exposed WT or *Apc*<sup>min/+</sup> organoid cells grown in 2D to *H. pylori* G27 for 6 h and then eradicated the bacteria with antibiotics for 24 or 48 h. Successful eradication was confirmed by plating (data not shown). Antibiotic eradication did not reverse the up-regulation of Lgr5 in either WT or *Apc*<sup>min/+</sup> organoid cells (Fig 6A–C), suggesting that Lgr5 up-regulation is a long-lived (up to 48 hour-long) consequence of *H. pylori* exposure, alone and especially in conjunction with Wnt pathway dysregulation. Maintaining organoid cells in 2D for up to 48 h without infection or antibiotics was *per se* not sufficient to increase their Lgr5 expression (Fig S4A and B). Western blotting for Lgr5 confirmed that Lgr5 expression is stable for at least 24 h post-eradication in WT organoids (Fig 6D and E). In line with the observed continued up-regulation of Lgr5 after successful eradication of the bacteria, treatment with antibiotics after *H. pylori* encapsulation into 3D cultures did not reverse the ability of *H. pylori*–exposed organoid cells to form spheroids (Fig 6F

and G). The results indicate that Lgr5 expression and stemness are comparatively long-term consequences of the interaction of *H. pylori* with its target cells that persist after antibiotic eradication of live bacteria.

## Discussion

A large body of evidence now points to a tight and consequential interaction of *H. pylori* with gastric stem and progenitor cells. Most studies have examined the antral Lgr5<sup>+</sup> stem and progenitor cell compartment, in both experimentally infected mice and patients presenting with *H. pylori* infection (Uehara et al, 2013; Sigal et al, 2015). A seminal study conducted in mice provided immunohistochemical evidence for *H. pylori* residing close to Lgr5<sup>+</sup> cells in the murine antrum (Sigal et al, 2015); this interaction with Lgr5<sup>+</sup> cells led to the chemotaxis- and T4SS-dependent expansion of the stem cell compartment and the induction of antimicrobial factors (Sigal et al, 2015). Later work confirmed that, indeed, Lgr5<sup>+</sup> gland base stem cells are the dominant cell type responding to the presence of the bacteria with NF-κB activation, as determined by nuclear accumulation of the acetylated form of the NF-κB subunit p65, and with pro-inflammatory cytokine production (Wizenty et al, 2022). Lgr5 itself appears to be dispensable for stem cell expansion and antimicrobial responses to *H. pylori*; rather, a signaling axis involving R-spondin three signaling via Lgr4 has been implicated in the *H. pylori*–induced proliferation of Lgr5<sup>+</sup> stem cells and in elevated NF-κB expression and activity in proliferative stem cells (Wizenty et al, 2022). Results from a variety of conditional mouse lines indicate that R-spondin 3/Lgr4 signaling synergizes with *H. pylori*–derived ADP heptose for full NF-κB activation in vivo (Wizenty et al, 2022). Our data are nicely in line with the observed in vivo findings; co-culture of *H. pylori* with organoid cells in 2D or 3D expands the Lgr5<sup>+</sup> stem cell pool and induces stemness properties in antral organoid cells. These consequences of *H. pylori* exposure are dependent on the ability of the bacteria to produce and translocate CagA and ADP heptose into their target cells and can be prevented by NF-κB inhibition. One limitation of our study was that the addition of ADP heptose alone was not sufficient to induce Lgr5 up-regulation, not even in conjunction with a T4SS mutant infection; this latter piece of evidence might suggest that CagA is more important than ADP heptose in driving Lgr5 expression and stemness properties.

Immunohistochemistry of human antral biopsies has confirmed that *H. pylori* reside close to mitotic cells at the proliferative zone of the gland base (Sigal et al, 2015) and are associated with an expansion of Lgr5<sup>+</sup> gland base stem cells (Uehara et al, 2013). The

---

**Figure 3. *Apc* inactivation enhances *H. pylori*–induced Lgr5 expression in both 2D and 3D gastric organoids.**
**(A, B, C, D, E)** WT and *Apc*<sup>min/+</sup> antrum-derived organoid cells were grown as monolayers and infected with *H. pylori* strain G27 (MOI of 50) for 6 h. Lgr5 expression was assessed by quantitative IF microscopy and Western blotting. **(A, B, C)** Representative IF images are shown in (A) (Lgr5 in magenta, β-tubulin in green, DAPI in blue; scale bar, 20 μm); the quantification of Lgr5 expression of a representative experiment is shown in (B) and means ± SD of three independent experiments are shown in (C) (plotted as fold change over control). **(D, E)** A representative Western blot is presented in (D) alongside the quantification (means ± SD) of three experiments in (E). **(F, G, H)** Antrum-derived organoids growing in Matrigel were infected with *H. pylori* strain G27 (MOI of 50) by encapsulation for 6 h and subjected to Western blotting of protein extracts and IF microscopy for the indicated markers. **(F)** Images of 3D organoids rendered in Fiji ImageJ using Maximum Intensity Z-Projection are shown in (F) (scale bar, 50 μm). **(G, H)** A representative Western blot is presented in (G) alongside the quantification (means ± SD) of three experiments in (H). **(B, C)** Data information: 700–1,200 cells per sample were analyzed in (B, C). **(B, C, E, H)** *P*-values were calculated using one-way ANOVA (non-parametric in (B); ordinary one-way ANOVA with Tukey's multiple comparisons test in (C, E, H)) and are indicated in ranges; \**P* <0.05; \*\*\**P* < 0.005; \*\*\*\**P* < 0.001; ns, not significant.

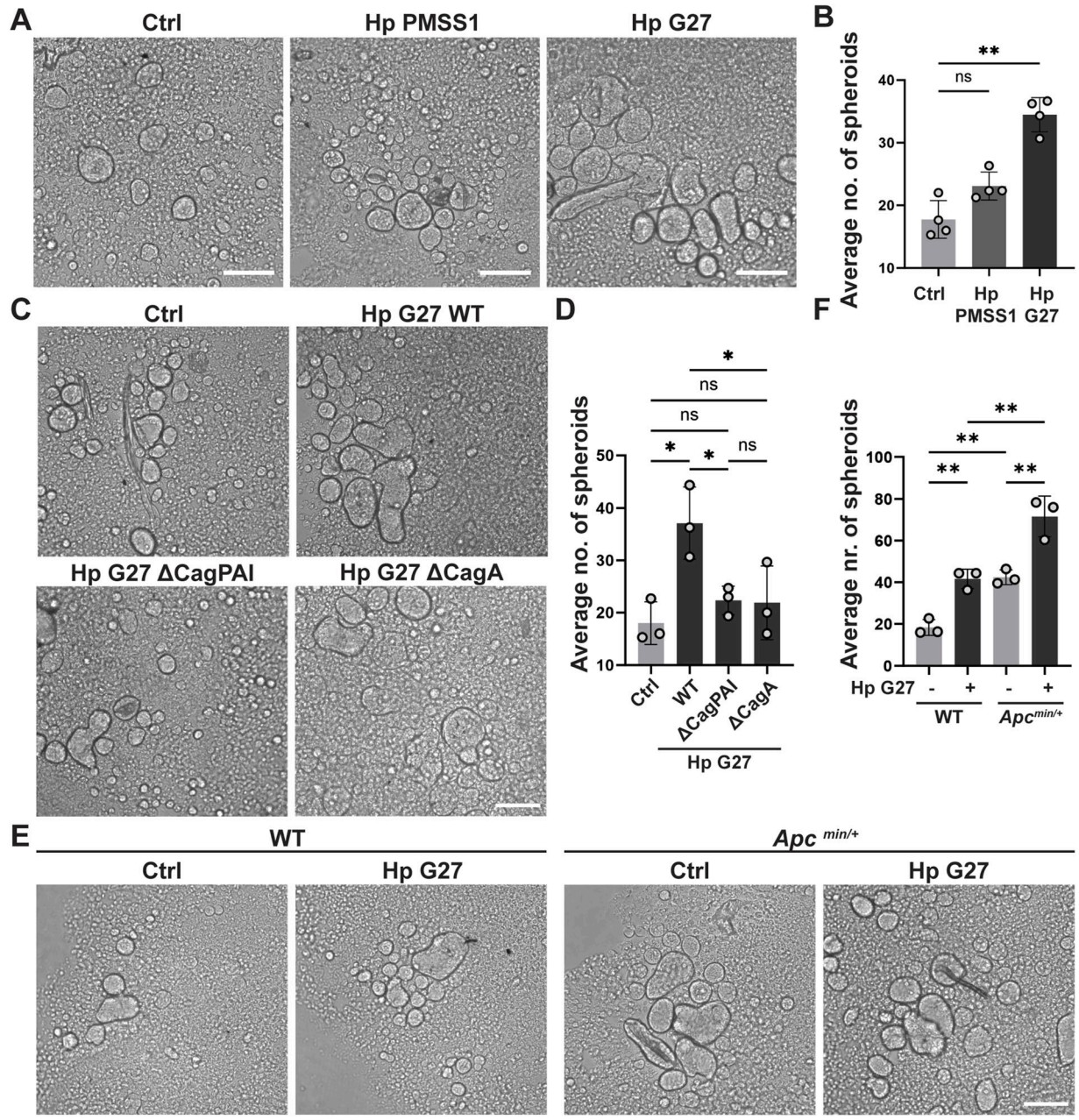

**Figure 4.  *H. pylori* induces stem cell-like properties as determined by spheroid formation assay.**
**(A, B)** WT antrum-derived organoids grown in Matrigel were exposed to *H. pylori* strains G27 or PMSS1 by encapsulation. After six hours, single-cell suspensions were generated and cells were re-seeded into anti-adhesive plates. **(A, B)** Representative images of spheroids are shown at 5 d post-seeding in (A) (scale bar, 125 μm) alongside the quantification (means ± SD) of four independent experiments in (B). **(C, D)** WT antrum-derived organoids were exposed to WT *H. pylori* G27 or its isogenic mutants by encapsulation. **(C, D)** Representative images of spheroids are shown at 5 d post-seeding in (C) (scale bar, 125 μm) alongside the quantification (means ± SD) of three experiments in (D). **(E, F)** WT and *Apc*^min/+ antrum-derived organoids were exposed to WT *H. pylori* G27 by encapsulation. **(E, F)** Representative images of spheroids are shown at 5 d post-seeding in (E) (scale bar, 125 μm) alongside the quantification (means ± SD) of three independent experiments in (F). Symbols represent biological replicates throughout. *P*-values were calculated by ordinary one-way ANOVA with Tukey's multiple comparisons test and are indicated in ranges; *$P$ <0.05; **$P$ < 0.01; ns, not significant.

latter was observed in *H. pylori*–positive gastritis without a diagnosis of gastric cancer and in non-neoplastic mucosa of *H. pylori*–positive gastric cancer patients but not in *H. pylori*–negative gastric cancer patients or *H. pylori*–negative healthy controls, suggesting a possible direct link between live *H. pylori* infection and

stem cell expansion (Uehara et al, 2013). A variety of additional studies have since confirmed an expansion of the Lgr5[+] stem cell pool in human *H. pylori*–positive gastritis or intestinal metaplasia, relative to *H. pylori*–negative normal gastric mucosa in geographically diverse patient cohorts (Levi et al, 2014; Choi et al, 2016;

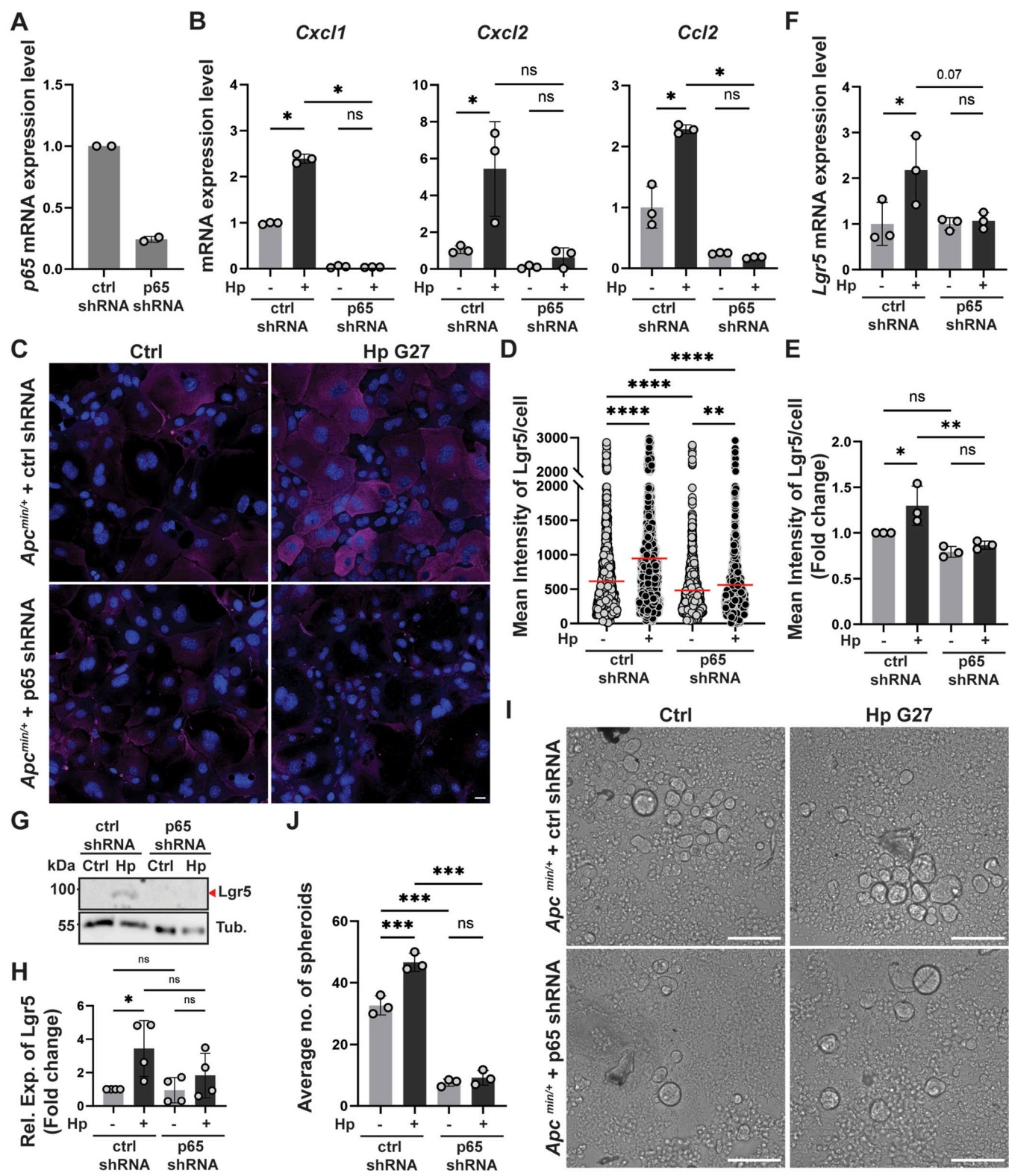

**Figure 5. The NF-κB subunit p65 is required for *H. pylori*–induced Lgr5 up-regulation and stemness.**
**(A, B, C, D, E, F, G, H)** *Apc*[min/+] antrum-derived organoid cells were engineered to constitutively express an shRNA specific for p65 or a scrambled shRNA. Cells of both genotypes were grown as monolayers and infected with *H. pylori* strain G27 (MOI of 50) for 6 h. **(A, B)** P65 knockdown efficiency as confirmed by qRT-PCR is shown in (A); the *H. pylori*–induced expression of the NF-κB target genes *Cxcl1*, *Cxcl2*, and *Ccl2* as examined by qRT-PCR is presented in (B). **(C, D, E)** Lgr5 expression was assessed by quantitative IF microscopy; representative IF images are shown in (C) (Lgr5 in magenta, DAPI in blue; scale bar, 20 μm) alongside the quantification of Lgr5 expression of a representative experiment in (D) and means ± SD of three independent experiments in (E) (plotted as fold change over control). **(F)** Lgr5 gene expression was quantified

Walker et al, 2018; Saberi et al, 2019). Elevated Lgr5 expression, especially in conjunction with other early gastric cancer markers such as CD44 and CD133, has been postulated to be predictive of a particularly high risk of progression to an advanced stage of disease, i.e., low-grade dysplasia or later (Choi et al, 2016; Walker et al, 2018).

We observe additive effects of constitutive Wnt/$\beta$-catenin signaling because of Apc truncation on the one hand and H. pylori infection on the other hand, on inducing Lgr5 expression and stemness properties. These data are reminiscent of an earlier study demonstrating an interaction between the Wnt/$\beta$-catenin and NF-$\kappa$B signaling pathways in a mouse model of intestinal stem cell hyperproliferation and tumorigenesis because of constitutive stable $\beta$-catenin expression in intestinal epithelial cells (Schwitalla et al, 2013). In this model, the genetic ablation of RelA/p65 in intestinal epithelial cells slowed crypt stem cell expansion and delayed tumorigenesis; this phenotype could be attributed to diminished DNA binding of $\beta$-catenin in the absence of RelA/p65 (Schwitalla et al, 2013). The reverse was also shown to be true as aberrantly active NF-$\kappa$B signaling because of genetic ablation of I$\kappa$B$\alpha$ shortened survival and induced dedifferentiation of non-stem cells (Schwitalla et al, 2013). Active NF-$\kappa$B signaling thus appears to synergize with Wnt signaling in inducing stem cell markers and stem cell-like properties; this may even occur by dedifferentiation of non-stem cells. We attempted live microscopy using organoid cells derived from our Lgr5-mOrange reporter mouse line to address whether H. pylori induces (more) Lgr5 expression in Lgr5$^+$ cells or de-differentiates Lgr5$^-$ cells but sadly found the endogenous mOrange reporter signal to be too weak and thus unsuitable for video microscopy.

Whether an expansion of the Lgr5$^+$ stem/progenitor cell pool is not only prognostically relevant as proposed (Choi et al, 2016; Walker et al, 2018) but also functionally significant remains to be experimentally proven. We and others have shown that antral Lgr5$^+$ stem/progenitor cells and their Troy$^+$ corpus counterparts are the targets of H. pylori–induced DNA damage in the form of DNA DSBs resulting from transcription/replication conflicts in actively cycling cells and in the form of oxidative damage of guanosine residues resulting from oxidative stress (Uehara et al, 2013; He et al, 2023). This evidence, together with genetic data showing that H. pylori synergizes with inherited mutations in HR repair genes in inducing gastric cancer (Usui et al, 2023) and experimental data showing that Lgr5$^+$ cells serve as cells of origin of gastric cancer in the antrum (Li et al, 2016; Fatehullah et al, 2021) and the corpus of the stomach (Leushacke et al, 2017), makes it seem plausible that an expansion of the stem cell pool by H. pylori is an early and critical event in gastric carcinogenesis. Further circumstantial evidence comes from the observation that T4SS-proficient strains (with the ability to induce DNA damage and stem cell expansion) are more tightly associated with gastric cancer or gastric precancerous lesions than T4SS-deficient strains, in both the human population (Huang et al, 2003) and experimentally infected animals (Rieder et al, 2005; Arnold et al, 2011). In summary, the combined data are consistent with the model that Lgr5$^+$ cell expansion is triggered by the bacterial metabolite ADP heptose and/or CagA via activation of the Alpk1/NF-$\kappa$B signaling axis, and serves as a protective response of the gland base to bacterial invasion; at the same time as stem cells or their secretory cell progeny produce antimicrobial factors, they are subject to DNA damage driven by transcription/replication conflicts, which may, if incorrectly or not at all repaired, constitute an early driver event in gastric malignant transformation.

## Materials and Methods

### Mice, organoid cultures, and H. pylori infections in 2D and 3D

C57BL/6J-Apc$^{Min}$/J mice (Strain #:002020) were obtained from the Jackson laboratory. Lgr5-2A-CreERT2-2A-mOrange2 mice were described previously (Fazilaty et al, 2021; Reichmuth et al, 2021 Preprint) and generously provided by Tomas Valenta and Konrad Basler. C57BL/6J WT and mutant mouse lines were maintained in groups of a maximum of five animals in individually ventilated cages at approved animal facilities of the University of Zurich, with access to food and water ad libitum. All mouse experimentation described in this study was approved by the Zurich Cantonal Veterinary Office (license no 35179, to A.M.) and complies with federal and cantonal regulations. Gastric organoid culture conditions were adapted based on previously published protocols (Bartfeld & Clevers, 2015); briefly, stomachs of 8–10 wk-old mice were harvested, opened along the greater curvature, and washed in ice-cold DPBS (#14190-094; Gibco). The mucus and muscle/serosa layer were removed under a stereomicroscope using forceps; the antrum and corpus were separated with a large safety margin, and both tissues were cut into 2–5 mm$^2$ sized pieces. Pieces were dissociated in 43.4 mM sucrose (#A2211,5000; Huberlab), 54.9 mM d-sorbitol (#240850; Sigma-Aldrich) in DPBS by 5–10 rounds of vigorous pipetting and washing, followed by incubation in 10 mM EDTA (#A1103,1000a; BioChemica) at 4°C for 1 h 20 min (rolling shaker). After one more washing step in 43.4 mM sucrose, 54.9 mM d-sorbitol, glands were extracted by placing tissue pieces under a glass slide (#AA00000112E01MNZ10; Epredia) and applying enough pressure to release individual glands; glands were collected in DMEM, pushed through a 40 $\mu$m filter, counted, collected by centrifuging, and mixed with Matrigel (#356231; Corning), seeded into 24-well plates (50 $\mu$l per well) and

---

by qRT-PCR (normalized to Gapdh) and is plotted as means ± SD of three independent experiments in (F). **(G, H)** A representative Western blot is presented in (G) alongside the quantification (means ± SD) of four experiments in (H). **(I, J)** Cells of both genotypes grown in Matrigel were exposed to H. pylori strain G27 by encapsulation. After six hours, single-cell suspensions were generated and cells were re-seeded into anti-adhesive plates. **(I, J)** Representative images of spheroids are shown at 5 d post-seeding in (I) (scale bar, 125 $\mu$m) alongside the quantification (means ± SD) of three independent experiments in (J). Symbols represent biological replicates throughout. P-values were calculated by ordinary one-way ANOVA with Tukey's multiple comparisons test and are indicated in ranges; *$P$ <0.05; **$P$ < 0.01; ***$P$ < 0.005; ****$P$ < 0.001; ns, not significant.

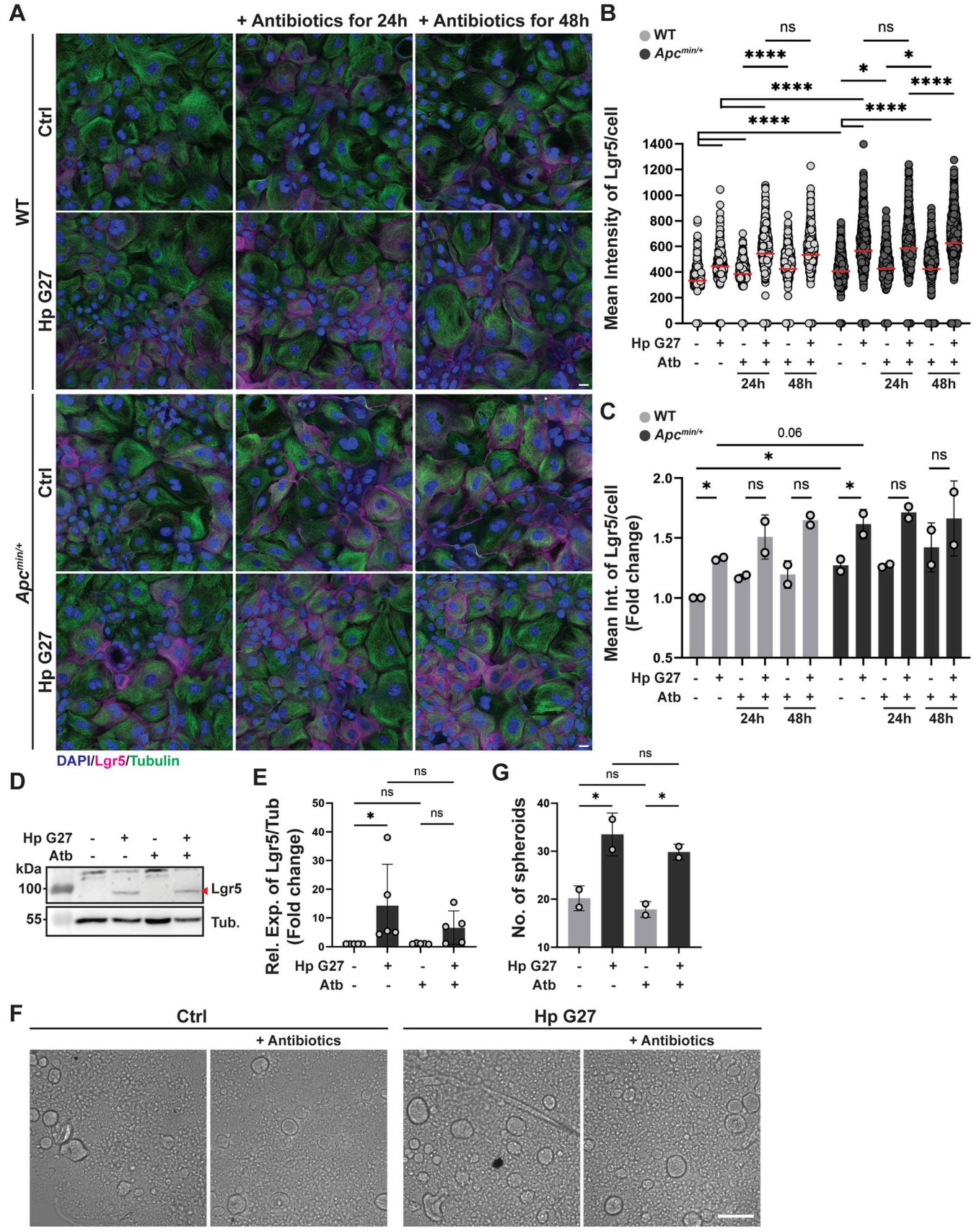

DAPI/Lgr5/Tubulin

cultivated at 37°C in ADMEM medium (Advanced DMEM/F12 [12634-010; Gibco] + 10 mM Pen/strep [#167369; Gibco]) supplemented with Wnt-conditioned medium (50% vol/vol; supernatants from L Wnt-3a cells; ATCC CRL-2647), R-spondin 1 (1 μg/ml; #315-32; PeproTech), N-acetylcysteine (1 mM, #A9165; Sigma-Aldrich) EGF (50 ng/ml, #AF-315-09; PeproTech), FGF (100 ng/ml, #100-26; PeproTech), Noggin (100 ng/ml, #250-38; PeproTech), [Leu15]-Gastrin I (10 nM, #G9145; Sigma-Aldrich) N2 (1x, #17502-048; Gibco), B27 (1x, #17504-044; Gibco), Hepes (10 mM, #15630-056; Gibco), GlutaMAX (2 mM, #35050-038; Gibco), Y27632 (10 μM; only on the seeding day; exclude thereafter, #S1049; Selleckchem) (referred to below as 3D organoid culture medium). For seeding in 2D, 7-d-old 3D organoids were dissociated in cold DMEM by vigorous pipetting. Organoid fragments were resuspended in TrypLE (#12605028; Gibco) and incubated at 37°C for 10 min. Single-cell suspensions were washed, centrifuged, counted, and seeded at 15,000 cells/well into rat tail collagen (#50201; Ibidi)-coated wells in 2D culture medium (which is identical to 3D medium but lacks Wnt-conditioned medium, R-spondin, Pen/Strep, FGF, Noggin, and N-acetylcysteine; contains 10% FCS, #10270-106; Gibco and 1 μM TGF-β inhibitor, #SML0788; Sigma-Aldrich); cells received fresh medium after 2 d in culture and were infected with H. pylori after 3 d in culture. H. pylori was grown as described previously (Sayi et al, 2009); bacterial numbers were determined by measuring the optical density at 600 nm, and bacteria were added to cells at an MOI of 50 for 6 h, unless otherwise specified. Bacteria were killed by addition of 1 μg/ml tetracycline for 24 h. The H. pylori strains used in this study were G27 and its isogenic ΔPAI and ΔCagA mutants (Censini et al, 1996) and PMSS1 and its isogenic ΔCagE and ΔRfaE mutants (Arnold et al, 2011). The isogenic PMSS1 ΔCagA mutant was kindly provided by Steffen Backert, University of Erlangen, Germany. Organoid cells grown in 2D were treated with the NF-κB inhibitor BAY 11-7082 (#B5556; Sigma-Aldrich) at 1 μM final concentration. β-ADP heptose (#tlrl-adph-I; InvivoGen) was added at 0.5 μM final concentration for 6 h. For the purpose of H. pylori injection in 3D, organoids were grown for at least 3 d without Pen/Strep before being individually injected using a 30G × ½″ insulin syringe filled with H. pylori in Brucella broth plus 10% FCS and 10 μg/ml vancomycin and incubation for 6 h at 37°C. For the purpose of H. pylori encapsulation into 3D organoids, murine antrum organoids were cultured in 3D without Pen/Strep for at least 3 d. Organoids cultured in 24-well plates in 50 μl Matrigel (#356231; Corning) were washed once with cold PBS, and non-enzymatically recovered by the addition of 500 μl of pre-chilled cell recovery solution (#354253; Corning) and rocking on ice for

30–45 min (rpm = 20). Organoids were further broken up into small pieces of 100–200 cells by gentle pipetting using a 1-ml tip (5–10 times). Pieces were collected in 19.5 ml of pre-chilled PBS by centrifugation at 350 g, 4°C for 5 min, resuspended in 500 μl of 3D organoid culture medium without Pen/Strep, transferred into a well of an ultra-low attachment surface 24-well plate (#3473; Corning) and supplemented with H. pylori grown in Brucella liquid culture medium (Sayi et al, 2009), usually at a (calculated) MOI of 250, and cultured for 6 h at 37°C. Reconstituted organoids were washed in 50 ml of PBS by centrifugation at 100 g, RT, for 5 min, and either resuspended directly in lysis buffer for Western blotting, in Matrigel for immunofluorescence microscopy or TrypLE for the spheroid formation assay.

## Engineering of organoid cells

To achieve p65 knockdown, an shRNA construct (forward sequence: CCGGCTGTCCTCTCACATCCGAT TTCTCGAGAAATCGGATGTGAGAGGACAGTTTTTG; reverse sequence: AATTCAAAAACTGTCCTCTCACAT CCGATTTCTCGAGAAATCGGATGTGAGAGGACAG) was inserted into the pLKO.1-TRC vector at the Age I and Eco RI restriction sites. For the control shRNA construct, the following oligonucleotides were used: forward sequence: CCGGCCTAAGGTTAAGTCGCCCTCGCTCGAGCGAGGGC-GACTTAACCTTAGGTTTTTG; reverse sequence: AATTCAAAAACCTAAGGT-TAAGTCGCCCTCGCTCGAGCGAGGGCGACTTAACCTTAGG. Packaging was carried out by transfecting 293 T cells with 12 μg of transfer plasmid, 6 μg of psPax2, and 3 μg of pCMV-VSVG in serum-free DMEM with polyethylenimine. After 3 d, virus-containing supernatants were harvested, filtered through a 0.45-μm filter, and concentrated by mixing three volumes of the supernatant with one volume of Lenti-X concentrator (Takara). This mixture was incubated overnight at 4°C and centrifuged at 1,500 g for 45 min at 4°C. The resulting virus pellets were resuspended in 250 μl of 3D organoid culture medium with polybrene (8 μg/ml). For transduction, organoids were dissociated by pipetting 30 times, followed by a 5-min incubation in TripLE (Gibco) at 37°C. The small cell clusters were collected by centrifugation, combined with 250 μl of the viral suspension, and subjected to spinoculation at 600 g and 32°C for 60 min. The plate was then incubated for 6 h at 37°C. After incubation, cells were centrifuged at 1,000 g for 5 min, resuspended in 100 μl of Matrigel (BD Biosciences), and divided into two wells of a 24-well culture plate. Once the Matrigel had polymerized, 500 μl of 3D organoid culture medium was added to each well. Puromycin selection was initiated 2 d after transduction (2 μg/ml for 3 d, followed by seven more days with puromycin of 0.6 μg/ml), followed by seeding of organoid cells in 2D for infection.

**Figure 6. Elevated Lgr5 expression and stem cell-like properties persist after eradication of *H. pylori*.**
**(A, B, C)** WT and *Apc*^min/+^ antrum-derived organoid cells were grown as monolayers and infected with *H. pylori* strain G27 (MOI of 50) for 6 h. Cells were then either fixed directly, or maintained in 100 IU/ml penicillin, 100 μg/ml streptomycin and 30 μg/ml tetracycline for 24 or 48 h before fixation. Lgr5 expression was assessed by quantitative IF microscopy. **(A, B, C)** Representative IF images are shown in (A) (Lgr5 in magenta, β-tubulin in green, DAPI in blue; scale bar, 20 μm); the quantification of Lgr5 expression of a representative experiment is shown in (B), and means ± SD of two independent experiments are shown in (C) (plotted as fold change over control). **(D, E)** Cells were seeded, infected, and treated with antibiotics as described above, and extracts were subjected to Lgr5-specific Western blotting. **(D, E)** A representative Western blot is presented in (D) alongside the quantification (means ± SD, plotted as fold change over control) of five experiments in (E). **(F, G)** WT antrum-derived organoids grown in Matrigel were exposed to *H. pylori* strain G27 by encapsulation. After six hours, single-cell suspensions were generated and cells were re-seeded into anti-adhesive plates, in the presence or absence of the three antibiotics listed above. **(F, G)** Representative images of spheroids are shown in (F) (scale bar, 125 μm) alongside the quantification of two independent experiments in (G). **(B, C)** Data information: 700–1,200 cells per sample were analyzed in (B, C). **(C, E, G)** Symbols in (B) represent individual cells; symbols in (C, E, G) represent biological replicates. **(B, C, E, G)** *P*-values were calculated using by one-way ANOVA (non-parametric in (B); ordinary one-way ANOVA with Tukey's multiple comparisons test in (C, E, G)) and are indicated in ranges; *$P$ <0.05; ****$P$ < 0.001; ns, not significant.

## Spheroid formation assay

We used Biofloat plates (#83.3925.400; Sarstedt) with anti-adhesive surface coating for quantitative spheroid formation assays. Single-cell suspensions were generated by incubation in TrypLE (#12605028; Gibco) at 37°C for 10 min, and single cells were seeded into Biofloat plates at seeding densities of 2,000–8,000 cells per well in 150 $\mu$l of 3D organoid medium. Plates were incubated at 37°C; spheroid formation was first visible between 24 and 48 h post-seeding. Spheroids with a cutoff diameter of >50 $\mu$m were counted under the microscope in triplicate on days 3, 5, 7, and 10; fresh growth medium was added every 3 d. Unless otherwise stated, data shown are from day 5 and a seeding density of 8,000 cells/well.

## Western blotting

Samples were run on 10% SDS–PAGE gels and transferred onto nitrocellulose membranes. The primary antibodies were anti-Lgr5 (#AB75850; Abcam) and anti–$\alpha$-tubulin (#T9026; Sigma-Aldrich), each used at a 1:500 dilution in TBS/Tween with 5% BSA. Blots were imaged using a Fusion Solo chemiluminescence imaging system (Vilber); signals were quantified using Fiji. The relative expression of Lgr5/$\alpha$-tubulin is shown throughout.

## Quantitative RT–PCR

Total RNA was isolated from 2D-cultured gastric organoids using the RNeasy extraction kit (QIAGEN). A total of 1,500 ng of RNA was reverse-transcribed using SuperScript III reverse transcriptase (Invitrogen). SYBR Green gene expression assays (Roche) were performed for the following genes using specific primers: *LGR5*, forward: GGCTCGGTGTGCTCCTGTCCT, reverse: TGCCTCAGGGAATG-CAGGCC; *Troy*, forward: TGTGTCCTCTGCAAACAGTGCG, reverse: CCAGTCTTCCTTGAACCGTTGC; *Aqp5*, forward: CGCTCAGCAACAACACA ACACC, reverse: GACCGACAAGCCAATGGATAAG; *Axin2*, forward: ATGGAGT CCCTCCTTACCGCAT, reverse: GTTCCACAGGCGTCATCTCCTT; *Ascl2*, forward: TCTCTCGGACCCTCTCTCAG, reverse: GGACCCCGTAC-CAGTCAAG; *Runx1*, forward: CACCTGTCTCTGCATCGCAGGACT, reverse: CCATCCGTGACAGATACGCACCTC; *Lrig1*, forward: TCTGCAGGAAGTG-TACCTCAACAG, reverse: GAGAG ACAACTCCTATGGAAGCAGT; *Sox2*, forward: AACGGCAGCTACAGCATGATGC, reverse: CGAGCTGGTCA TGGAGTTGTAC; *Bmi1*, forward: CAGGTTCACAAAACCAGACCAC, reverse: TGACGGGTGAGCTGCATAAA. The samples were analyzed using a LightCycler 480 instrument (Roche), with target mRNA levels calculated relative to Gapdh.

## Immunofluorescence microscopy

For immunofluorescence microscopy of organoid cell mono-layers, cells were seeded into $\mu$-Slide eight Well chambered coverslips with ibiTreat surface modification (#80806; Ibidi). Cells were infected with bacteria as described above for 6 h, and washed twice with 1x PBS to remove unattached bacteria and debris. Cells were fixed with 4% PFA/1x PBS for 30 min when gently rocking (10–15 rpm, Heidolph DUOMAX 1030 shaker). After a brief wash with 1x PBS, cells were permeabilized with 0.1% Triton X-100/1x PBS for 30 min and blocked with blocking

solution (1% Serum, 200 mM NH$_4$Cl in 1x PBS) for 1 h. Cells were then incubated with primary antibodies diluted in blocking solution for 2 h as follows: anti-LGR5 mouse monoclonal (#MA5-25644, 1:100; Invitrogen), anti–$\beta$-tubulin rat monoclonal (#ab6160; 1:200; Abcam), and anti–*H. pylori* rabbit polyclonal (#B0471; 1:50; DAKO). Cells were then washed three times with 1x PBS, followed by incubation with secondary antibodies diluted in blocking solution for 1 h. Secondary antibodies were Alexa Fluor Plus 488 goat anti-mouse IgG (#A32723; 1:500; Thermo Fisher Scientific), Alexa Fluor 568 goat anti-rabbit IgG (#A11036; 1:500; Thermo Fisher Scientific), and Alexa Fluor Plus 647 goat anti-rat (#A48265; 1:500; Thermo Fisher Scientific). After three brief washes with 1x PBS, cells were stained with 10 $\mu$g/ml DAPI diluted in 1x PBS for 15 min and stored in 1x PBS at 4°C. High-throughput imaging was performed using the high-content imaging system ImageXpress Confocal HT.ai (Molecular Devices) equipped with motorized stage, a CMOS sensor and dual spinning disk tech-nology. Images were acquired using 50-micron pinhole disk and a 40x (NA = 1.15) Nikon objective with water immersion. 25–36 images with z-planes spanning range up to 20 $\mu$m were acquired for each well. The images were then rendered in Fiji software using a Maximum Intensity Z-Projection (MIP). Quantitative image-based analysis (QIBC) was performed using CellProfiler v4.2.4. For the analysis, the DAPI signal was used for nuclei segmentation and $\beta$-tubulin was used for cell segmentation. The "Cell cytoplasm" was then generated using CellProfiler plugins and the Lgr5 signal was quantified for each cell cytoplasm. ~1,000 cells were quantified per condition. To this end, 35–40 high-powered (40x) fields were randomly chosen and all cells in those fields were analyzed. For immunofluorescence microscopy of 3D organoids, organoids infected or not by microinjection or en-capsulation as described above were embedded in Matrigel and grown in 24-well plates with glass bottom (#P24-1.5H-N; Cellvis) in 3D medium for 24 h. Organoids were washed with 1x PBS for 15 min, fixed with 4% PFA/1x PBS for 1 h, permeabilized with 0.5% Triton X-100 for 1 h, blocked with blocking solution (2% serum/1x PBS supplemented with 0.1% Triton X-100) for 1 h, and stained with the antibodies listed above diluted in blocking solution for 2 h (4 h for organoids encapsulated with *H. pylori*) at RT. All incubation steps were performed with gentle rocking (10–15 rpm). Organoids were then washed three times with 1x PBS for 15 min, followed by incubation with secondary antibodies (same as above) diluted at 1:400 (1:300 for organoids encap-sulated with *H. pylori*) in blocking solution for 1 h. After staining, organoids were washed three times with 1x PBS for 15 min and stored in 1x PBS at 4°C. Nuclei were stained with 50 $\mu$g/ml DAPI diluted in 1x PBS for 30 min before imaging. High-throughput imaging was performed with an automated super resolution system from Olympus (Olympus IXplore SpinSR10), equipped with the spinning disk from Yokogawa (CSU-W1), motorized Z-drift compensation (IX3-ZDC2), two sCMOS cameras, and a 20x (NA = 0.8) Olympus objective and/or 30x (NA = 1.05) Olympus objective with a silicon oil immersion. For each well, a low-resolution image was acquired to cover the complete well us-ing a 4x (NA = 0.16) Olympus objective. This overview image was used to localize the coordinates of specific positions which were then imaged with high resolution. For each site, z-planes

spanning range up to 500 *μm* were acquired. The images were then rendered in Fiji software using a MIP. For the analysis of 3D data, MIPs were generated for each acquired z-stack field, later being stitched together to obtain a view for each channel and organoid. Organoids were then segmented using CellProfiler module. The watershed algorithm was then applied on a binary mask generated by Otsu thresholding. For each segmented organoid, nuclei and cell cytoplasm were identified as individual objects, and features quantifying intensities for each channel were extracted per object.

### Statistical analyses

All statistical analyses were performed using GraphPad Prism software. Non-parametric one-way ANOVA (Kruskal–Wallis) was used for the statistical analysis of individual experiments with >500 data points per condition. Ordinary one-way ANOVA with Tukey's multiple comparisons test was used for statistical analyses of comparisons of replicate experiments of three or more groups/conditions. $*P < 0.05$; $**P < 0.01$; $***P < 0.005$; and $****P < 0.001$.

## Data and Materials Availability

All data are available in the main text or the supplementary materials.

## Supplementary Information

## Acknowledgements

We thank all members of the Müller laboratory for helpful discussions. This work was supported by the Swiss National Science Foundation project grant 310030_192490 (A Müller), the Forschungskredit of the University of Zürich (J He and Z Nascakova) and the Medical Faculty of the University of Zürich (A Müller).

### Author Contributions

Z Nascakova: conceptualization, funding acquisition, data curation, formal analysis, investigation, visualization, methodology, and writing—review and editing.
J He: conceptualization, funding acquisition, data curation, formal analysis, investigation, visualization, methodology, and writing—review and editing.
G Papa: data curation, formal analysis, investigation, visualization, methodology, and writing—review and editing.
B Francas: investigation, methodology, and writing—review and editing.
F Azizi: data curation, formal analysis, investigation, methodology, and writing—review and editing.
A Müller: conceptualization, supervision, funding acquisition, project administration, and writing—original draft, review, and editing.

### Conflict of Interest Statement

The authors declare that they have no conflict of interest.

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
