## [Reviewer comments · Life Science Alliance]

Life Science Alliance

Helicobacter pylori induces the expression of Lgr5 and stem cell properties in gastric target cells

Zuzana Nascakova, Jiazhuo He, Giovanni Papa, Biel Francas, Flora Azizi, and Anne Müller

DOI: <https://doi.org/10.26508/lsa.202402783>

Corresponding author(s): Anne Müller, University of Zurich

Review Timeline:

Submission Date:	2024-04-22
Editorial Decision:	2024-05-28
Revision Received:	2024-08-15
Editorial Decision:	2024-08-16
Revision Received:	2024-08-19
Accepted:	2024-08-19

Transaction Report:

May 28, 2024

Re: Life Science Alliance manuscript #LSA-2024-02783-T

Prof. Anne Müller
University of Zürich
Institute of Molecular Cancer Research
Winterthurerstr. 190
Zürich 8057
SWITZERLAND

Dear Dr. Müller,

Thank you for submitting your manuscript entitled "Helicobacter pylori induces the expression of Lgr5 and stem cell properties in gastric target cells" to Life Science Alliance. The manuscript was assessed by expert reviewers, whose comments are appended to this letter. We invite you to submit a revised manuscript addressing the Reviewer comments.

Thank you for this interesting contribution to Life Science Alliance. We are looking forward to receiving your revised manuscript.

Sincerely,

B. MANUSCRIPT ORGANIZATION AND FORMATTING:

Reviewer #1 (Comments to the Authors (Required)):

Using an organoid infection model, the authors demonstrate in this study an effect of *H. pylori* on Lgr5 expression and stemness of gastric epithelial cells.

This is a nice study that provides interesting new insights into the ability of *H. pylori* to interfere with gastric stemness. Some of the data confirm previous studies in mice and organoids, while some aspects are novel. Most of the data are rather of descriptive nature, the mechanistic exploration of the phenotype is rather limited. Nevertheless, from my point of view this study is interesting and elegantly performed, thus being a good fit to this journal

Specific comments:

Intro:

"the third most common cause of cancer-related deaths (Bray et al, 2018)" -> see update
<https://acsjournals.onlinelibrary.wiley.com/doi/10.3322/caac.21660>: fourth most common cause

- "such experimentally induced cancers can be driven by KrasG12D expression alone" (Leushacke et al., 2017) -> Leushacke found metaplasia, but no cancer

- "we asked how *H. pylori* would affect the biology of this lineage. In particular, we asked whether, and how, *H. pylori* increases the stem cell pool and induces stem cell properties in its target cells" because this has been addressed extensively before (e.g. Sigal 2015) maybe specify "...in organoids" or similar.

Results/discussion:

- Fig. 1A-C: Is the Lgr5 signal homogenously intracellular? Quantification of Lgr5+ cells missing.

- Fig. 1E: HP G27 quantification more than 60% of tubulin signal (image in D shows less)

- Fig. 1F: DAPI signal seems over-exposed

- Fig. 1B,G: statistics should be done by comparing averages per biological replicate, not the individual cells, please also adjust for all other figures where this applies

- Fig. 1G: Is *H. pylori* in contact with Lgr5+ cells? Is Lgr5 increased in areas where *H. pylori* accumulates? Please state that expansion medium (full medium) was used.

- How does Lgr5- and other Stem cell gene expression change upon infection? E.g. Ascl2?

- Fig. 2A, F: Lgr5+ cells quantification missing, Lgr5 RNA data would add value....also for the other experiments

- Fig. EV2A,B: addition of ADP hep is not sufficient to induce Lgr5 -> Why not? dRfaE as shown has no Lgr5 expression, but adding the only factor missing in that mice is not effective? What happens if deltaCAgE infection plus ADP Helptose are used?

- Fig. 2I: NFKB p65 or NFKB target genes expression in the respective conditions would show that NFKB is active in the organoid model (and also if it is active in the Lgr5+ cells?) and demonstrate efficacy of the NFKB inhibitor. Uninfected controls treated with NFKBi would add value.

- Fig. 3A-H: synergistic or additive effect of APC Mut + Infection? Both terms used in this chapter. Significance bar for WT+ vs. APC+ seem to be not accurately placed.

What is the mechanism for this synergism? Should be at least discussed..

- Fig. 5A,B Antibiotics not only not reverse Lgr5 upregulation, but indeed lead to more Lgr5 - controls without antibiotics but same time +24h and +48h are missing. Maybe Lgr5 is even higher at later time points (than 6 hours).

- Discussion: The experiment where addition of ADP hep does not increase Lgr5 should be discussed?

Reviewer #2 (Comments to the Authors (Required)):

This manuscript describes a series of experiments performed to investigate if *H. pylori* increases the stem cell pool - defined as Lgr5+ cells through T4SS activity - using organoid models. There is broad interest in understanding mechanisms of increased proliferation and stimulation of stem cells in cancers and in particular in *H. pylori*- associated cancer.

It is much appreciated that for their methodology they explored Lgr5 expression in a few capacities including IF, Western, and reporter mouse. In 2D and 3D organoids and western blotting. Further, their discussion around previous findings in the antrum and in relationship to Rspo/Lgr4 findings previously reported by other groups are well addressed. Their study lacks complemented bacterial strains (not always possible in *H. pylori*).

The overall conclusion is that *H. pylori* does induce expression of Lgr5 - a stem cell marker - if the strain has a functional T4SS and further, that ApcMin mutation exacerbates the impact of *H. pylori* on Lgr5 expression. Finally, inhibition of NFkB can also reduce Lgr5 expression in these co-cultures.

Major considerations:

The abstract and introduction speak specifically of cancers of the antrum. Please address the location of gastric cancers in humans and mouse models to help the reader understand the significance of cancers in the antrum v. corpus.

What is the dominant band in the Lgr5 western blots that is between 100 and 130kDa?

Clarify how the expression of Lgr5 was quantified. Was it all cells in a number of fields? Recognizing that 600-1000 cells per condition were analyzed its not clear how those cells were chosen - all cells in the fields until that many cells were counted? In many of the IF images it appears that there is or isn't Lgr5 expression after *H. pylori* co-culture. Is it possible to know if the cells which are Lgr5+ are the same cells that have T4SS activation? And/or are they particular specialized epithelial cell type that is found in these organoids? (do they also express Mist or other markers)

The data presentation is not completely transparent Re: replicates and data presentation standards.

1. Connecting lines between experimental groups or conditions in line graphs should not be done (usually reserved for paired analysis).

2. Presentations of averages is helpful but the standard deviation is not presented in these graphs.

3. There is no indication if the western blots are performed as biological or technical replicates and there are no error bars or statistical analysis performed.

4. There is no indication if the spheroids are performed as biological or technical replicates and there are no error bars or statistical analysis performed.

Some conclusions are not fully justified without statistical analyses.

Minor concerns which should be addressed:

In the final sentence of the introduction the meaning of "*H. pylori* synergized with hyperactivation of the Wnt signaling pathway" is not clear based on the short description of the findings. Clarify.

Results section 1. The authors suggest they only make organoids from antral tissue (no line numbers but first long sentence) but they present results from both corpus-derived and antral derived tissues. (Figure 1 and EV1B). Please be accurate.

The authors suggest this antibody clone is "validated for this application" but they don't describe validation in the methods and materials. (or cite this statement in the results) Please rectify.

Please provide in the results a brief description of the MOI of infection- this is a vital piece. Provide rationale for use of MOI of 250 for some assays.

What is the name of the PMSS1 mutant mouse referred to in this line "A PMSS1 mutant lacking the ability to produce and deliver the LPS biosynthetic intermediate -ADP-heptose was completely deficient for Lgr5 induction (Fig. EV2A,B)." Please add it to the text.

The rationale for only performing organoid experiments on ApcMin/+ organoids with strain G27 is not clear (for Lgr5 expression assays and for bacterial eradication studies). Does PMSS1 produce the same results?

It is not clear that the antibiotic usage in the 3D cultures results in an impact on *H. pylori* survival. Please confirm that "encapsulated" *H. pylori* are killed in the cultures. Bacterial components are still present but the T4SS would not be 'active'. I am not sure how much this adds new knowledge to the spheroid formation. What if the organoids were split again for example? The timelines are not immediately clear from reading the text either. (what is persistent, long-lived time line)

Reviewer #3 (Comments to the Authors (Required)):

"*Helicobacter pylori* induces expression of Lgr5 and stem cell properties in gastric target cells" is a nice study that uses murine gastric-derived organoids to assess Lgr5 expression in response to *H. pylori*. The authors demonstrate that *H. pylori* increases Lgr5 in gastric organoids derived from the antrum and corpus and is dependent on a functional type IV secretion system and NF-kB signaling. Further, they demonstrate that antibiotic treatment does not reverse *H. pylori*-induced increases in Lgr5+ cells. Although, it has been established that *H. pylori* increases the number of stem cells and Lgr5+ stem cell proliferation in a type IV

secretion system-dependent manner in vivo, this manuscript optimizes the gastric organoid model to interrogate other factors involved in *H. pylori*-induced Lgr5 expression.

The following are suggestions for the authors' consideration:

- 1) Gastric cancer is now the 5th most common cause of cancer related death (Bray et al. 2024), Introduction, page 3.
- 2) Lgr5 expression is very robust via IF, but quite low or undetectable by Western blots. What accounts for this discrepancy in Lgr5 expression? What controls were used to show specificity of Lgr5 staining by IF?
- 3) Since Lgr5 is so robust by IF, on average what percentage of cells are staining positive for Lgr5 in response to *H. pylori* under the conditions tested?
- 4) For Lgr5 Western blots, why are Lgr5 levels undetectable for uninfected cells and in some cases extremely low with *H. pylori* infection. The authors observe an *H. pylori* dose-dependent response in Lgr5 by IF (Figure EV1A). Is the same observed by Western blot? Would higher MOIs allow for better detection of Lgr5 by Western blot?
- 5) How does the mean intensity of Lgr5/cell compare between delta CagE, delta CagA, and delta RfaE (Figure 2 and EV2)? Include Lgr5 IF images for delta RfaE mutant compared to controls (Figure EV2).
- 6) Include IF for NF-kB β experiments (Figure 2I-J). Did the authors also stain for NF-kB in these experiments to show NF-kB activation with *H. pylori* and the mutants and decreased nuclear localization with the inhibitor? Is there colocalization of nuclear NF-kB and Lgr5+ cells in response to *H. pylori*?
- 7) There is inconsistency in the number of replicates for each experiment, within some experiments only performed once (Figure 2G-H, EV2 C-D, EV3D). N=3 should be included for all experiments for quantification of data and appropriate statistical tests.
- 8) Western blot data is quantified as relative expression of Lgr5/Tubulin, but plotted on an XY graph with connected data points. These data might be better represented as a scatter dot plot or column plot with standard error in order to perform the appropriate statistical analyses.
- 9) It appears that *H. pylori* increases the average number of spheroids, in a cagE- and cagA-dependent manner (Figure 4), what effect does deletion of RfaE have on the number of spheroids? Is the spheroid formation directly dependent on Lgr5-mediated proliferation?
- 10) Some *H. pylori* strains are able to induce higher levels of Lgr5, can this be correlated with the known virulence factors assessed in this study (T4SS formation, CagA translocation, levels of ADP heptose) or the level of inflammation and injury caused by these respective strains in vivo?

Point-by-point Response to Reviewers

We thank our reviewers for constructive criticism and for sharing great ideas for improving this work. We have now addressed all reviewer comments, mostly with additional experimentation, as specified in the point-by-point response below (answers in red).

Reviewer #1 (Comments to the Authors (Required)):

Using an organoid infection model, the authors demonstrate in this study an effect of *H. pylori* on Lgr5 expression and stemness of gastric epithelial cells.

This is a nice study that provides interesting new insights into the ability of *H. pylori* to interfere with gastric stemness.

Some of the data confirm previous studies in mice and organoids, while some aspects are novel. Most of the data are rather of descriptive nature, the mechanistic exploration of the phenotype is rather limited. Nevertheless, from my point of view this study is interesting and elegantly performed, thus being a good fit to this journal

Specific comments:

Intro:

"the third most common cause of cancer-related deaths (Bray et al, 2018)" -> see update <https://acsjournals.onlinelibrary.wiley.com/doi/10.3322/caac.21660>: fourth most common cause

It is now, according to the latest data, the fifth most common cause of cancer-related death. This has been changed, and the latest reference has been added.

Introduction, p.3: The causal link to cancer is especially well documented for *H. pylori* and gastric cancer (Plummer *et al*, 2015), the fifth most common cause of cancer-related deaths (Bray *et al*, 2024)...

- "such experimentally induced cancers can be driven by KrasG12D expression alone" (Leushacke *et al.*, 2017) -> Leushacke found metaplasia, but no cancer

True. This has now also been corrected:

Introduction, p.4: Gastric Lgr5⁺ stem cells not only regenerate gastric glands at steady state and following tissue damage, respectively, but, upon oncogenic insult, may serve as cells-of-origin of advanced gastric cancer in mice; such experimentally induced gastric cancers, or their precursor lesion "spasmolytic polypeptide-expressing metaplasia" (SPEM), can be driven by Kras^{G12D} expression alone (Leushacke *et al.*, 2017) or in conjunction with *Apc* and *Trp53* ablation (Fatehullah *et al*, 2021)....

- "we asked how *H. pylori* would affect the biology of this lineage. In particular, we asked whether, and how, *H. pylori* increases the stem cell pool and induces stem cell properties in its target cells" because this has been addressed extensively before (e.g. Sigal 2015) maybe specify "...in organoids" or similar.

This sentence has been modified as follows to clarify that organoids were used as the model system here:

Introduction, p.5: ...We set out to investigate, using organoid cells, whether and how *H. pylori* would increase the stem cell pool and induce stem cell properties in its target cells. Infection of mice, and the immunohistochemical evaluation of human biopsies, had previously suggested that the stem and progenitor compartment expands during *H. pylori* infection (Sigal *et al*, 2015; Uehara *et al*, 2013)...

Results/discussion:

- Fig. 1A-C: Is the Lgr5 signal homogenously intracellular? Quantification of Lgr5+ cells missing.

We feel that the signal is mostly, but not exclusively at the membrane. In stainings where antibodies against tubulin and Lgr5 are combined, it is possible to observe an Lgr5 ring around the cytoplasmic tubulin signal (in most cells). Cells with a very strong Lgr5 signal show it everywhere. We are now including the frequency of Lgr5⁺ cells in figures where this analysis is appropriate. This way of looking at the data has confirmed the trends previously reported as “mean intensity of Lgr5”. Both plots are now shown side by side in Figures 1 and 2 and their supplements. The text has been modified as follows:

Results, p.6: Exposure to G27, and to a lesser extent PMSS1, resulted in enhanced Lgr5 expression as determined by immunofluorescence microscopy using an antibody clone that had been validated for this application using organoids from an *Lgr5*-mOrange reporter mouse (He *et al.*, 2023), followed by automated signal quantification (Fig 1A-D); both the mean intensity of the Lgr5 signal per cell, and the frequency of Lgr5⁺ cells increased due to infection (Fig1A-D)....

Figure legend to Figure 1, p.23: ...The mean Lgr5 intensity across all cells per condition of four independently conducted experiments is plotted as mean +/- SD of the fold change over (uninfected) control in C. The frequencies of Lgr5⁺ cells (i.e. with a mean intensity > the mean of the control condition) among all cells, of the four experiments in C are shown in D; symbols in C and D represent independent experiments. ...

Results, p.7. In the G27 strain background, a mutant lacking the entire Cag pathogenicity island encoding the T4SS was completely defective for Lgr5 induction as determined by microscopy of cells infected in 2D, both with respect to the mean Lgr5 intensity per cell, and the frequency of Lgr5⁺ cells (Fig 2A-D).

Results, p.7: This was indeed the case for both strains using an inhibitor of the $\kappa\text{B}\alpha$ kinase IKK, which prevents $\kappa\text{B}\alpha$ phosphorylation and the nuclear translocation of NF- κB ; exposure to this compound reduced both the mean Lgr5 intensity per cell, and the frequency of Lgr5⁺ cells (Fig S2G-J). The combined results...

- Fig. 1E: HP G27 quantification more than 60% of tubulin signal (image in D shows less)

The quantification of WBs has been redone throughout (there was indeed a mistake in the way the data were plotted, thank you for picking this up). We have now quantified three or more WBs in panels showing WB data, and are showing Lgr5 expression (normalized to tubulin) as fold change over control throughout. Note that only one representative WB (instead of all three) is now shown in Figure 1F (former Figure 1E). We had to make space for qRT-PCR data, which also nicely show that Lgr5 transcript is induced by *H. pylori* exposure of organoid cells.

Results, p.6: An increase in the expression of Lgr5 could also be detected at the transcript level by qRT-PCR (Fig 1E). Western blotting further also confirmed the upregulation of Lgr5, not only in

antrum-derived cultures (Fig 1F,G), but also in corpus-derived cultures in which the Lgr5 signal was generally weaker (Fig S1B).

- Fig. 1F: DAPI signal seems over-exposed

We have reduced the DAPI intensity in Figure 1F (now H).

- Fig. 1B,G: statistics should be done by comparing averages per biological replicate, not the individual cells, please also adjust for all other figures where this applies

Every organoid culture is a bit different in terms of its baseline Lgr5 expression, which complicates statistical analyses across experiments. We have now solved this problem by calculating the fold change in mean intensity over the control condition; this has allowed us to properly statistically analyse all experiments. We are now showing both a representative experiment and data from three or more replicate experiments (with statistics) in consecutive figure panels throughout (as well as the frequencies as requested above).

- Fig. 1G: Is *H. pylori* in contact with Lgr5⁺ cells? Is Lgr5 increased in areas where *H. pylori* accumulates? Please state that expansion medium (full medium) was used.

H. pylori is definitely in contact with Lgr5⁺ cells, but not exclusively. We are now stating this as we call out suppl. Figure 1A, and are also mentioning in the figure legend that 2D culture medium (different from expansion medium) was used.

Results, p.6: ...Lgr5 expression increased in a multiplicity of infection (MOI)-dependent manner; *H. pylori* was found to adhere to both Lgr5⁺ and Lgr5⁻ cells in the culture (Fig S1A).

Figure legends, p.26: **A-G**, Antrum-derived wild type organoid cells seeded as monolayers in 2D culture medium were infected...

- How does Lgr5- and other Stem cell gene expression change upon infection? E.g. Ascl2?

This is an interesting question. We have now measured the expression of 9 different gastric stem cell markers (Troy, Sox2, Runx1, Ascl2 etc in addition to Lgr5) in three independent experiments, and none of them, except for Lgr5 and Ascl2, are induced upon infection. This is now shown in suppl. Figure 1 and explained in the text as follows:

Results, p.6: The induction of Lgr5 was fairly specific, as other gastric stem cell markers, such as Sox2, Troy, Runx1, Lrig1, and others (Liabeuf *et al*, 2022) were not induced as a consequence of infection (Fig S1C); an exception was Ascl2, the expression of which mirrored that of Lgr5 (Fig S1C).

- Fig. 2A, F: Lgr5+ cells quantification missing, Lgr5 RNA data would add value....also for the other experiments

We have now added the frequencies of Lgr5⁺ cells as new panels in Figure 2. The text has been modified as follows:

Results, p.7: In the G27 strain background, a mutant lacking the entire Cag pathogenicity island encoding the T4SS was completely defective for Lgr5 induction as determined by microscopy of cells infected in 2D, both with respect to the mean Lgr5 intensity per cell, and the frequency of Lgr5⁺ cells

(Fig 2A-D). A mutant lacking only CagA showed an intermediate phenotype and had some residual Lgr5-inducing activity (Fig 2A-D). The same was true for the PMSS1 strain background, in which T4SS deficiency (in the Δ CagE mutant) completely abolished the strain's Lgr5-inducing activity whereas lack of CagA again showed an intermediate phenotype (Fig 2E-H).

Lgr5 RNA data indeed also adds value, not only in Figure 1 (see above), but also here, where G27 WT but not its mutants, induce Lgr5 transcription:

Results, p.7: This could be confirmed for G27 also at the transcript level (Fig 2I).

- Fig. EV2A,B: addition of ADP hep is not sufficient to induce Lgr5 -> Why not? dRfaE as shown has no Lgr5 expression, but adding the only factor missing in that mice is not effective? What happens if deltaCagE infection plus ADP Heptose are used?

We also don't quite understand this discrepancy. It may very well be that CagA is more important than ADP-heptose in inducing the Lgr5 phenotype.

We have now done the interesting experiment suggested by this reviewer, ie. we have infected cells with the Δ PAI mutant of G27 and added ADP-Heptose on top. This treatment did not rescue the phenotype of the mutant in three independent experiments, which is stated in the text as follows (please also see below for the discussion of the inability of ADP-hep to act alone in inducing Lgr5, last point raised by reviewer):

Results, p.7: Addition of β -ADP-heptose alone was not sufficient to induce Lgr5 (Fig S2A-C), also not when combined with a Δ PAI mutant infection (Fig S2D-F).

- Fig. 2I: NFKB p65 or NFKB target genes expression in the respective conditions would show that NFKB is active in the organoid model (and also if it is active in the Lgr5+ cells?) and demonstrate efficacy of the NFKB inhibitor. Uninfected controls treated with NFKBi would add value.

Unfortunately, all our attempts (using a whole battery of antibodies) at staining NF- κ B P65 together with Lgr5 failed. To overcome the liabilities of using a small molecule inhibitor of NF- κ B, we generated P65kd organoids and infected them with *H. pylori*, followed by Lgr5 microscopy, Western blotting, qRT-PCR and spheroid formation assay. This series of experiments confirmed that NF- κ B is absolutely required for all observed phenotypes; the data are included in a new figure (Figure 5), alongside the following text:

Results, p.10: **RNAi-driven knockdown of the NF- κ B subunit P65 reverses the effects of *H. pylori* infection on Lgr5 expression and stemness**

In order to confirm the critical role of NF- κ B in *H. pylori*-induced Lgr5 expression and stemness, we generated organoid cells with a constitutive overexpression of an shRNA targeting the NF- κ B subunit p65. The knockdown of p65 was >75 % as judged by qRT-PCR (Fig 5A) and *H. pylori*-induced NF- κ B target gene expression (of *Cxcl1*, *Cxcl2* and *Ccl2*) was strongly reduced (Fig 5B). Lgr5-specific immunofluorescence microscopy revealed that the p65-specific shRNA, but not the scrambled control shRNA, prevented Lgr5 induction by strain G27 (Fig 5C-E). This was confirmed by Lgr5-specific qRT-PCR (Fig 5F) and Western blotting (Fig 5G,H). Loss of p65 also compromised the ability of *H. pylori* G27 to induce spheroid formation; in fact, p65 knock-down cells had a much lower capacity to form spheroids already at baseline (Fig 5I,J), despite similar growth kinetics of the organoids. The combined results suggest that *H. pylori*-induced Lgr5 expression and enhanced stemness is dependent on NF- κ B.

- Fig. 3A-H: synergistic or additive effect of APC Mut + Infection? Both terms used in this chapter. Significance bar for WT+ vs. APC+ seem to be not accurately placed. What is the mechanism for this synergism? Should be at least discussed..

The bar has been repositioned, and we are sticking to the term “additive” or something neutral (“exacerbated”) throughout (probably more accurate). There is not much literature reporting synergy between constitutive b-catenin activation and the NF-κB pathway. One paper showing synergy between the two pathways is discussed extensively, as follows:

Discussion, p. 12: This data is reminiscent of an earlier study demonstrating an interaction between the Wnt/β-catenin and NF-κB signaling pathways in a mouse model of intestinal stem cell hyperproliferation and tumorigenesis due to constitutive stable β-catenin expression in intestinal epithelial cells (Schwitalla *et al*, 2013). In this model, the genetic ablation of RelA/p65 in intestinal epithelial cells slowed crypt stem cell expansion and delayed tumorigenesis; this phenotype could be attributed to diminished DNA binding of β-catenin in the absence of RelA/p65 (Schwitalla *et al.*, 2013). The reverse was also shown to be true as aberrantly active NF-κB signaling due to genetic ablation of IκBα shortened survival and induced dedifferentiation of non-stem cells (Schwitalla *et al.*, 2013). Active NF-κB signaling thus appears to synergize with Wnt signaling in inducing stem cell markers and stem-cell-like properties; this may even occur by de-differentiation of non-stem cells.

- Fig. 5A,B Antibiotics not only not reverse Lgr5 upregulation, but indeed lead to more Lgr5 - controls without antibiotics but same time +24h and +48h are missing. Maybe Lgr5 is even higher at later time points (than 6 hours).

We have now performed stainings for these later control cells, and don't see an increase over time alone. The new data have been added to the supplement of this figure, with the following changes to the text:

Results, p.10: Antibiotic eradication did not reverse the upregulation of Lgr5 in either WT or *Apc*^{min/+} organoid cells (Fig 6A-C), suggesting that Lgr5 upregulation is a persistent and long-lived consequence of *H. pylori* exposure, alone and especially in conjunction with Wnt pathway dysregulation.

Maintaining organoid cells in 2D for up to 48 hours without infection or antibiotics was *per se* not sufficient to increase their Lgr5 expression (Fig S4A,B).

-

- Discussion: The experiment where addition of ADP hep does not increase Lgr5 should be discussed?

This is now mentioned as follows:

Discussion, p.12: These consequences of *H. pylori* exposure are dependent on the ability of the bacteria to produce and to translocate CagA and ADP-heptose into their target cells, and can be prevented by NF-κB inhibition. One limitation of our study was that the addition of ADP-heptose alone was not sufficient to induce Lgr5 upregulation, not even in conjunction with a T4SS mutant infection; this latter piece of evidence might suggest that CagA is more important than ADP-heptose in driving Lgr5 expression and stemness properties.

Reviewer #2 (Comments to the Authors (Required)):

This manuscript describes a series of experiments performed to investigate if *H. pylori* increases the stem cell pool - defined as Lgr5+ cells through T4SS activity - using organoid models. There is broad interest in understanding mechanisms of increased proliferation and stimulation of stem cells in cancers and in particular in *H. pylori*- associated cancer.

It is much appreciated that for their methodology they explored Lgr5 expression in a few capacities including IF, Western, and reporter mouse. In 2D and 3D organoids and western blotting. Further, their discussion around previous findings in the antrum and in relationship to Rspo/Lgr4 findings previously reported by other groups are well addressed. Their study lacks complemented bacterial strains (not always possible in *H. pylori*).

The overall conclusion is that *H. pylori* does induce expression of Lgr5 - a stem cell marker - if the strain has a functional T4SS and further, that ApcMin mutation exacerbates the impact of *H. pylori* on Lgr5 expression. Finally, inhibition of NFkB can also reduce Lgr5 expression in these co-cultures.

Major considerations:

The abstract and introduction speak specifically of cancers of the antrum. Please address the location of gastric cancers in humans and mouse models to help the reader understand the significance of cancers in the antrum v. corpus.

We have inserted a sentence and two additional references speaking about the site of gastric carcinogenesis (driven by Lgr5+ cancer stem cells) in humans and mice, as follows:

Introduction, p.4: LGR5 expression is a hallmark not only of murine, but also of human gastric cancers (Leushacke M et al, 2017). Lgr5 marks cancer stem cells that give rise to both corpus and antrum/pyloric gastric cancer in mice (Fatehullah A et al, 2021, Leushacke M et al, 2017, Tan SH et al, 2020), recapitulating the dominant gastric sites at which human non-cardia gastric cancer arises (Kim SJ & Choi CW, 2019).

What is the dominant band in the Lgr5 western blots that is between 100 and 130kDa?

This unspecific band is always there. We don't know what it is (its strength does not correlate with the strength of the Lgr5 signal).

Clarify how the expression of Lgr5 was quantified. Was it all cells in a number of fields? Recognizing that 600-1000 cells per condition were analyzed its not clear how those cells were chosen - all cells in the fields until that many cells were counted?

Yes, it was all cells in 35-40 high-powered fields. This is now specified in more detail as follows:

Materials and Methods, p.20: The "Cell cytoplasm" was then generated using Cellprofiler plugins and the Lgr5 signal was quantified for each cell cytoplasm. ~1000 cells were quantified per condition. To this end, 35 to 40 high-powered (40x) fields were randomly chosen and all cells in those fields were analyzed....

In many of the IF images it appears that there is or isn't Lgr5 expression after *H. pylori* co-culture. Is it possible to know if the cells which are Lgr5+ are the same cells that have T4SS activation? And/or are they particular specialized epithelial cell type that is found in these organoids? (do they also express Mist or other markers)

This is something we have wondered about as well. Unfortunately, all our attempts (using a whole battery of antibodies) at staining nuclear NF- κ B P65 (as a marker of T4SS activation) together with Lgr5 failed. To overcome the liabilities of using a small molecule inhibitor of NF- κ B, we generated P65kd organoids and infected them with *H. pylori*, followed by Lgr5 microscopy, Western blotting, qRT-PCR and spheroid formation assay. This series of experiments confirmed that NF- κ B is absolutely required for all observed phenotypes; the data are included in a new figure (Figure 5), alongside the following text:

Results, p.10: RNAi-driven knockdown of the NF- κ B subunit P65 reverses the effects of *H. pylori* infection on Lgr5 expression and stemness

In order to confirm the critical role of NF- κ B in *H. pylori*-induced Lgr5 expression and stemness, we generated organoid cells with a constitutive overexpression of an shRNA targeting the NF- κ B subunit p65. The knockdown of p65 was >75 % as judged by qRT-PCR (Fig 5A) and *H. pylori*-induced NF- κ B target gene expression (of *Cxcl1*, *Cxcl2* and *Ccl2*) was strongly reduced (Fig 5B). Lgr5-specific immunofluorescence microscopy revealed that the p65-specific shRNA, but not the scrambled control shRNA, prevented Lgr5 induction by strain G27 (Fig 5C-E). This was confirmed by Lgr5-specific qRT-PCR (Fig 5F) and Western blotting (Fig 5G,H). Loss of p65 also compromised the ability of *H. pylori* G27 to induce spheroid formation; in fact, p65 knock-down cells had a much lower capacity to form spheroids already at baseline (Fig 5I,J), despite similar growth kinetics of the organoids. The combined results suggest that *H. pylori*-induced Lgr5 expression and enhanced stemness is dependent on NF- κ B.

We have also now looked at 8 additional stem cell markers in 3 independent experiments, to see if any of them might be co-regulated with Lgr5. This was not the case for most of them, with the exception of *Ascl2*, which shows a pattern that parallels that of Lgr5. This is now shown in suppl. Figure 1 and explained in the text as follows:

Results, p.6: The induction of Lgr5 was fairly specific, as other gastric stem cell markers, such as Sox2, Troy, Runx1, Lrig1, and others (Liabeuf *et al*, 2022) were not induced as a consequence of infection (Fig S1C); an exception was *Ascl2*, the expression of which mirrored that of Lgr5 (Fig S1C).

The data presentation is not completely transparent Re: replicates and data presentation standards.

1. Connecting lines between experimental groups or conditions in line graphs should not be done (usually reserved for paired analysis).

We have removed the connecting lines and are now showing means +/- SD throughout. As the baseline expression of Lgr5 differed somewhat from one experiment to the next, we have resorted to plotting fold changes relative to the untreated control throughout.

2. Presentations of averages is helpful but the standard deviation is not presented in these graphs.

This has been corrected; means +/-SD are now shown throughout.

3. There is no indication if the western blots are performed as biological or technical replicates and there are no error bars or statistical analysis performed.

All Western blots for which we are showing quantifications were performed as biological replicates (ie. they represent different, independently conducted experiments). This is now clarified in the figure legends throughout.

E.g. figure legend of Figure 1, p.26: ...A representative Western blot is presented in D (Lgr5, ~100 kDa, indicated by red arrows, and α -tubulin, ~50 kDa) alongside the quantification of three independent experiments (plotted as mean \pm SD, normalized to tubulin and the control condition) in G.

4. There is no indication if the spheroids are performed as biological or technical replicates and there are no error bars or statistical analysis performed.

This has also been corrected. Except for some accessory data (pilot study etc) in the supplemental figures, where this is now clearly stated, we are showing biological replicate experiments throughout. We are now specifying in the figure legends that symbols represent biological replicates.

E.g. figure legend, p.28: Wild type and *Apc*^{min/+} antrum-derived organoids were exposed to wild-type *H. pylori* G27 by encapsulation. Representative images of spheroids are shown at five days post seeding in E (scale bar, 125 μ m) alongside the quantification (means \pm SD) of three independent experiments in G. Symbols represent biological replicates throughout.

Some conclusions are not fully justified without statistical analyses.

Minor concerns which should be addressed:

In the final sentence of the introduction the meaning of "H. pylori synergized with hyperactivation of the Wnt signaling pathway" is not clear based on the short description of the findings. Clarify.

This paragraph has been updated and rephrased as follows for clarity:

Introduction, p. 5: We show here, using organoid cells infected with *H. pylori* in 2D or in 3D, that *H. pylori* induces the stem cell marker Lgr5 in its target cells and that exposure to T4SS-proficient, but not -deficient *H. pylori* results in enhanced stemness properties. Hyperactivation of the Wnt signaling pathway by inactivation of the tumor suppressor *Apc* exacerbated the *H. pylori*-induced stemness of target cells. Lgr5 induction and stemness could further be prevented completely by the pharmacological inhibition or genetic inactivation of NF- κ B, and was stable even after antibiotic eradication of the bacteria.

Results section 1. The authors suggest they only make organoids from antral tissue (no line numbers but first long sentence) but they present results from both corpus-derived and antral derived tissues. (Figure 1 and EV1B). Please be accurate.

This has been corrected:

Results, p.6: To assess whether *H. pylori* infection affects Lgr5 expression, we cultured organoids from antrum or corpus tissue harvested from C57BL/6 mice, seeded the organoid cells in 2D and exposed them to the *H. pylori* strains PMSS1 or G27 for 6 hours; both strains harbor a functional T4SS and adhere to murine cells in culture (He J et al, 2023).

The authors suggest this antibody clone is "validated for this application" but they don't describe validation in the methods and materials. (or cite this statement in the results) Please rectify.

We had previously validated the antibody for a recent publication, and are now citing this properly as follows:

Results, p.6: Exposure to G27, and to a lesser extent PMSS1, resulted in enhanced Lgr5 expression as determined by immunofluorescence microscopy using an antibody clone that had been validated for this application using organoids from an *Lgr5-mOrange* reporter mouse (He J et al, 2023), followed by automated signal quantification (Fig 1A-D); ...

Please provide in the results a brief description of the MOI of infection- this is a vital piece. Provide rationale for use of MOI of 250 for some assays.

This information is now provided as follows:

Results, p.6: ...both the mean intensity of the Lgr5 signal per cell, and the frequency of Lgr5⁺ cells increased due to infection (Fig 1A-D). Lgr5 expression increased in a multiplicity of infection (MOI)-dependent manner; as an MOI of 50 showed strong effects on Lgr5 expression without completely overgrowing the cells or causing cytotoxicity (Fig. S1A, and data not shown), this MOI was used throughout the study unless otherwise indicated.

A (theoretical) MOI of 250 is only used in encapsulation experiments, to make sure that at least a few live bacteria end up inside the organoids. The staining for Hp (e.g. in suppl Figure 1F) shows that the number of bacteria in the organoid lumen is actually exceedingly low. The actual MOI is probably not higher than 1.

Suppl Figure legend, Figure S1: ...Note that, although a theoretical MOI of 250 was used for encapsulation, the actual MOI as estimated by *H. pylori*-specific staining is probably as low as 1. ...

What is the name of the PMSS1 mutant mouse referred to in this line "A PMSS1 mutant lacking the ability to produce and deliver the LPS biosynthetic intermediate β -ADP-heptose was completely deficient for Lgr5 induction (Fig. EV2A,B)." Please add it to the text.

This has been corrected:

Results, p.7: A PMSS1 mutant lacking the ability to produce and deliver the LPS biosynthetic intermediate β -ADP-heptose (Δ RfaE) was completely deficient for Lgr5 induction (Fig S2A-C).

The rationale for only performing organoid experiments on ApcMin/+ organoids with strain G27 is not clear (for Lgr5 expression assays and for bacterial eradication studies). Does PMSS1 produce the same results?

G27 is by far the more potent strain when it comes to Lgr5 and stemness induction. Therefore, in the second half of the manuscript, we predominantly show G27 data in the main figures; some PMSS1 data is included in the supplemental information, but this was never as clear as data generated with G27.

It is not clear that the antibiotic usage in the 3D cultures results in an impact on *H. pylori* survival. Please confirm that "encapsulated" *H. pylori* are killed in the cultures. Bacterial components are still present but the T4SS would not be 'active'. I am not sure how much this adds new knowledge to the spheroid formation. What if the organoids were split again for example? The timelines are not immediately clear from reading the text either. (what is persistent, long-lived time line)

This is an important point. We of course confirmed that the antibiotics killed the *H. pylori*. This, and the timelines, are now specified more concretely as follows:

Results, p.10: To this end, we exposed WT or *Apc*^{min/+} organoid cells grown in 2D to *H. pylori* G27 for six hours, and then eradicated the bacteria with antibiotics for 24 or 48 hours. Successful eradication was confirmed by plating (data not shown). Antibiotic eradication did not reverse the upregulation of Lgr5 in either WT or *Apc*^{min/+} organoid cells (Fig 6A-C), suggesting that Lgr5 upregulation is a long-lived (up to 48 hour-long) consequence of *H. pylori* exposure, alone and especially in conjunction with Wnt pathway dysregulation. Maintaining organoid cells in 2D for up to 48 hours without infection or antibiotics was *per se* not sufficient to increase their Lgr5 expression (Fig S4A,B). Western blotting for Lgr5 confirmed that Lgr5 expression is stable for at least 24 hours post eradication in WT organoids (Fig 6D,E). In line with the observed continued upregulation of Lgr5 after successful eradication of the bacteria, treatment with antibiotics after *H. pylori* encapsulation into 3D cultures did not reverse the ability of *H. pylori*-exposed organoid cells to form spheroids (Fig 6F,G). The results indicate that Lgr5 expression and stemness are comparatively long-term consequences of the interaction of *H. pylori* with its target cells that persist after antibiotic eradication of live bacteria.

Reviewer #3 (Comments to the Authors (Required)):

"Helicobacter pylori induces expression of Lgr5 and stem cell properties in gastric target cells" is a nice study that uses murine gastric-derived organoids to assess Lgr5 expression in response to *H. pylori*. The authors demonstrate that *H. pylori* increases Lgr5 in gastric organoids derived from the antrum and corpus and is dependent on a functional type IV secretion system and NF-κB signaling. Further, they demonstrate that antibiotic treatment does not reverse *H. pylori*-induced increases in Lgr5+ cells. Although, it has been established that *H. pylori* increases the number of stem cells and Lgr5+ stem cell proliferation in a type IV secretion system-dependent manner in vivo, this manuscript optimizes the gastric organoid model to interrogate other factors involved in *H. pylori*-induced Lgr5 expression.

The following are suggestions for the authors' consideration:

1) Gastric cancer is now the 5th most common cause of cancer related death (Bray et al. 2024), Introduction, page 3.

This has been corrected, with the following modification made to the text:

Introduction, p.3: The causal link to cancer is especially well documented for *H. pylori* and gastric cancer (Plummer et al, 2015), the fifth most common cause of cancer-related deaths (Bray et al, 2024)...

2) Lgr5 expression is very robust via IF, but quite low or undetectable by Western blots. What

accounts for this discrepancy in Lgr5 expression? What controls were used to show specificity of Lgr5 staining by IF?

The main control used to show specificity of the Lgr5 staining is the Lgr5 reporter mouse. More than 90% of Lgr5-mOrange reporter cells were stained by our antibody, and 66% of cells stained by the antibody also expressed the reporter. This data was already shown in our study He, Nascakova et al. Science Advances 2023, and is now referenced here when we first show Lgr5 staining using the same antibody. The text has been amended as follows:

Results, p.6: Exposure to G27, and to a lesser extent PMSS1, resulted in enhanced Lgr5 expression as determined by immunofluorescence microscopy using an antibody clone that had been validated for this application using organoids from an *Lgr5*-mOrange reporter mouse (He *et al.*, 2023), followed by automated signal quantification (Fig 1A-D); both the mean intensity of the Lgr5 signal per cell, and the frequency of Lgr5⁺ cells increased due to infection (Fig 1A-D).

Please note that we are now also showing qRT-PCR data to confirm at the transcript level that Lgr5 is induced by *H. pylori*. The data are included in Figures 1, 2 and 5, and described in the text as follows:

Results, p.6: Lgr5 expression increased in a multiplicity of infection (MOI)-dependent manner; *H. pylori* was found to adhere to both Lgr5⁺ and Lgr5⁻ cells in the culture (Fig S1A). An increase in the expression of Lgr5 could also be detected at the transcript level by qRT-PCR (Fig 1E).

Results, p.7: The same was true for the PMSS1 strain background, in which T4SS deficiency (in the Δ CagE mutant) completely abolished the strain's Lgr5-inducing activity whereas lack of CagA again showed an intermediate phenotype (Fig 2E-H). This could be confirmed for G27 also at the transcript level (Fig. 2I).

Results, p.10: Lgr5-specific immunofluorescence microscopy revealed that the p65-specific shRNA, but not the scrambled control shRNA, prevented Lgr5 induction by strain G27 (Fig 5C-E). This was confirmed by Lgr5-specific qRT-PCR (Fig 5F) and Western blotting (Fig 5G,H).

3) Since Lgr5 is so robust by IF, on average what percentage of cells are staining positive for Lgr5 in response to *H. pylori* under the conditions tested?

The frequencies of Lgr5⁺ cells are now shown alongside the mean intensities per cell, at least in the first two figures, with modifications to the text as follows:

Results, p.6: Exposure to G27, and to a lesser extent PMSS1, resulted in enhanced Lgr5 expression as determined by immunofluorescence microscopy using an antibody clone that had been validated for this application using organoids from an *Lgr5*-mOrange reporter mouse (He *et al.*, 2023), followed by automated signal quantification (Fig 1A-D); both the mean intensity of the Lgr5 signal per cell, and the frequency of Lgr5⁺ cells increased due to infection (Fig 1A-D).

Results, p.7: In the G27 strain background, a mutant lacking the entire Cag pathogenicity island encoding the T4SS was completely defective for Lgr5 induction as determined by microscopy of cells infected in 2D, both with respect to the mean Lgr5 intensity per cell, and the frequency of Lgr5⁺ cells (Fig 2A-D).

Results, p.7: This was indeed the case for both strains using an inhibitor of the I κ B α kinase IKK, which prevents I κ B α phosphorylation and the nuclear translocation of NF- κ B; exposure to this

compound reduced both the mean Lgr5 intensity per cell, and the frequency of Lgr5⁺ cells (Fig S2G-J). The combined results...

4) For Lgr5 Western blots, why are Lgr5 levels undetectable for uninfected cells and in some cases extremely low with *H. pylori* infection. The authors observe an *H. pylori* dose-dependent response in Lgr5 by IF (Figure EV1A). Is the same observed by Western blot? Would higher MOIs allow for better detection of Lgr5 by Western blot?

It is true that there is a certain discrepancy between the WB and IF data. All WB data were already generated with an MOI of 50 (the highest of the dose response shown for IF), and we were reluctant to go even higher. The MOI is now included in the figure legend for clarity throughout.

5) How does the mean intensity of Lgr5/cell compare between delta CagE, delta CagA, and delta RfaE (Figure 2 and EV2)? Include Lgr5 IF images for delta RfaE mutant compared to controls (Figure EV2).

Representative images are now included in suppl Figure 2A to illustrate the effect of RfaE deletion on Lgr5 expression. We are now showing fold changes of mean intensities of Lgr5 expression/cell of the infected over the control condition throughout, which facilitates comparisons within and across experiments. Our impression is that inactivation of RfaE and of CagE have very comparable effects in the PMSS1 background, as both essentially fail completely to induce Lgr5.

6) Include IF for NF-κBi experiments (Figure 2I-J). Did the authors also stain for NF-κB in these experiments to show NF-κB activation with *H. pylori* and the mutants and decreased nuclear localization with the inhibitor? Is there colocalization of nuclear NF-κB and Lgr5⁺ cells in response to *H. pylori*?

Representative images are now also included for NF-κB inhibitor experiments as requested (suppl. Figure 2G).

Unfortunately, all our attempts (using a whole battery of antibodies) at staining NF-κB P65 together with Lgr5 failed. To be certain that NF-κB is involved in Lgr5 expression upon *H. pylori* infection, we generated P65kd organoids and infected them with *H. pylori*, followed by Lgr5 microscopy, Western blotting, qRT-PCR and spheroid formation assay. This series of experiments confirmed that NF-κB is absolutely required for all observed phenotypes; the data are included in a new figure (Figure 5), alongside the following text:

Results, p.9: RNAi-driven knockdown of the NF-κB subunit P65 reverses the effects of *H. pylori* infection on Lgr5 expression and stemness

In order to confirm the critical role of NF-κB in *H. pylori*-induced Lgr5 expression and stemness, we generated organoid cells with a constitutive overexpression of an shRNA targeting the NF-κB subunit p65. The knockdown of p65 was >75 % as judged by qRT-PCR (Fig 5A) and *H. pylori*-induced NF-κB target gene expression (of *Cxcl1*, *Cxcl2* and *Ccl2*) was strongly reduced (Fig 5B). Lgr5-specific immunofluorescence microscopy revealed that the p65-specific shRNA, but not the scrambled control shRNA, prevented Lgr5 induction by strain G27 (Fig 5C-E). This was confirmed by Lgr5-specific qRT-PCR (Fig 5F) and Western blotting (Fig 5G,H). Loss of p65 also compromised the ability of *H. pylori* G27 to induce spheroid formation; in fact, p65 knock-down cells had a much lower capacity to form spheroids already at baseline (Fig 5I,J), despite similar growth kinetics of the

organoids. The combined results suggest that *H. pylori*-induced Lgr5 expression and enhanced stemness is dependent on NF- κ B.

7) There is inconsistency in the number of replicates for each experiment, within some experiments only performed once (Figure 2G-H, EV2 C-D, EV3D). N=3 should be included for all experiments for quantification of data and appropriate statistical tests.

We have repeated numerous experiments and now show N=3 for almost all of the experiments in main figures (the Lgr5 reporter experiment is an exception, this was done only 2 times; the antibiotic exposure was also done only twice with an IF readout, but 5 times with a Western blot readout). Every organoid culture is a bit different in terms of its baseline Lgr5 expression, which complicates statistical analyses across experiments. We have now attempted to solve this problem by calculating the fold change in mean intensity over the control condition; this has allowed us to properly statistically analyze all experiments. We are now showing both a representative experiment and data from three or more replicate experiments (with statistics) in consecutive figure panels throughout.

A statistics section has been added to the Materials and Methods section:

M&M, p. 21: **Statistical analyses.** All statistical analyses were performed using GraphPad Prism software. Non-parametric one-way ANOVA (Kruskal-Wallis) was used for the statistical analysis of individual experiments with >500 datapoints per condition. Ordinary one-way ANOVA with Tukey's multiple comparisons test was used for statistical analyses of comparisons of replicate experiments of three or more groups/conditions. *, p<0.05; **, p<0.01; *** p<0.005; **** p<0.001.

8) Western blot data is quantified as relative expression of Lgr5/Tubulin, but plotted on an XY graph with connected data points. These data might be better represented as a scatter dot plot or column plot with standard error in order to perform the appropriate statistical analyses.

Thank you for this suggestion. All WB data (and all IF data) are now shown as mean +/- SD (fold change over control) and statistical testing is done throughout.

9) It appears that *H. pylori* increases the average number of spheroids, in a cagE- and cagA-dependent manner (Figure 4), what effect does deletion of RfaE have on the number of spheroids? Is the spheroid formation directly dependent on Lgr5-mediated proliferation?

We have an RfaE mutant only in the PMSS1 background, which is a comparatively poor inducer of spheroid formation. Given the tricky interpretation of PMSS1 spheroid formation data, we have decided to move all PMSS1 spheroid assay data to the supplement, and made sure to avoid overinterpretation of the results in the text. We felt given the suboptimal induction of spheroid formation with WT bacteria, it was not worth it to go for the mutant. I hope our reviewer agrees with us on this matter.

It would be great to know whether spheroid formation depends on Lgr5. We have begun with the cloning of an inducible vector for the CRISPR/Cas9 knockout of Lgr5 (and of Lgr4), but this is not ready for lentivirus production, let alone for experiments. It will be another 3-4 months until this data is available, and we felt it was somewhat outside of the scope of this work.

10) Some *H. pylori* strains are able to induce higher levels of Lgr5, can this be correlated with the known virulence factors assessed in this study (T4SS formation, CagA translocation, levels of ADP heptose) or the level of inflammation and injury caused by these respective strains in vivo?

Such an analysis would require a larger collection of strains, and an in depth knowledge of their virulence properties. Ironically, despite its ability to colonize mice, PMSS1 is not particularly good at adhering to murine cells in vivo (G27 does it much better). We think that the lack of adherence underlies the generally weaker phenotypes observed with *H. pylori*, but this is only speculation at this point. We will look at more strains and their ability to induce Lgr5 and stemness in the future, but this is a new project in itself.

August 16, 2024

RE: Life Science Alliance Manuscript #LSA-2024-02783-TR

Prof. Anne Müller
University of Zurich
Institute of Molecular Cancer Research
Winterthurerstr. 190
Zurich 8057
Switzerland

Dear Dr. Müller,

Thank you for submitting your revised manuscript entitled "Helicobacter pylori induces the expression of Lgr5 and stem cell properties in gastric target cells". We would be happy to publish your paper in Life Science Alliance pending final revisions necessary to meet our formatting guidelines.

- please be sure that the authorship listing and order is correct
- please add a category for your manuscript to our system
- please make sure that the author order in the manuscript matches with the order in our system
- please add a figure callout for Figure S2 K,L to your main manuscript text
- under the "Mice, organoid cultures and H. pylori infections in 2D and 3D" section of the Materials and Methods, please explicitly indicate approval for the performed animal work, and who granted the approval.

A. FINAL FILES:

B. MANUSCRIPT ORGANIZATION AND FORMATTING:

Sincerely,

August 19, 2024

RE: Life Science Alliance Manuscript #LSA-2024-02783-TRR

Prof. Anne Müller
University of Zurich
Institute of Molecular Cancer Research
Winterthurerstr. 190
Zurich 8057
Switzerland

Dear Dr. Müller,

Thank you for submitting your Research Article entitled "Helicobacter pylori induces the expression of Lgr5 and stem cell properties in gastric target cells". It is a pleasure to let you know that your manuscript is now accepted for publication in Life Science Alliance. Congratulations on this interesting work.

DISTRIBUTION OF MATERIALS:

Again, congratulations on a very nice paper. I hope you found the review process to be constructive and are pleased with how the manuscript was handled editorially. We look forward to future exciting submissions from your lab.

Sincerely,
